# Re-definition of claudin-low as a breast cancer phenotype

Christian Fougner [1,2], Helga Bergholtz [1,2], Jens Henrik Norum [1] & Therese Sørlie [1,2 ✉]

The claudin-low breast cancer subtype is defined by gene expression characteristics and encompasses a remarkably diverse range of breast tumors. Here, we investigate genomic, transcriptomic, and clinical features of claudin-low breast tumors. We show that claudin-low is not simply a subtype analogous to the intrinsic subtypes (basal-like, HER2-enriched, luminal A, luminal B and normal-like) as previously portrayed, but is a complex additional phenotype which may permeate breast tumors of various intrinsic subtypes. Claudin-low tumors are distinguished by low genomic instability, mutational burden and proliferation levels, and high levels of immune and stromal cell infiltration. In other aspects, claudin-low tumors reflect characteristics of their intrinsic subtype. Finally, we explore an alternative method for identifying claudin-low tumors and thereby uncover potential weaknesses in the established claudin-low classifier. In sum, these findings elucidate the heterogeneity in claudin-low breast tumors, and substantiate a re-definition of claudin-low as a cancer phenotype.

[1] Department of Cancer Genetics, Oslo University Hospital, Oslo, Norway. [2] Institute for Clinical Medicine, University of Oslo, Oslo, Norway. ✉email: therese.sorlie@rr-research.no

The five breast cancer intrinsic subtypes were initially identified by hierarchical clustering of genes with significantly greater variation in expression between different breast tumors than between paired tumor samples pre- and post-chemotherapy[1,2]. Claudin-low breast tumors did not emerge as an independent group in this analysis. The claudin-low breast cancer subtype was discovered 7 years later in an integrated analysis of human and murine mammary tumors[3]. The existence of this subtype has later been observed in several independent breast cancer cohorts[4–9], and an analogous claudin-low subtype has been identified in bladder cancer[10,11].

The claudin-low breast cancer subtype is defined by gene expression characteristics, most prominently: Low expression of cell–cell adhesion genes, high expression of epithelial–mesenchymal transition (EMT) genes, and stem cell-like/less differentiated gene expression patterns[12]. Beyond these gene expression features, claudin-low tumors have marked immune and stromal cell infiltration[9,12], but are in many other aspects remarkably heterogeneous. No specific genomic aberrations accurately delineate claudin-low tumors, and there is a greater variation in mutational burden and degree of copy number aberration (CNA) than in the other breast cancer subtypes[13]. Claudin-low tumors are, however, often genomically stable, potentially due to their less differentiated state and a protective effect mediated by the EMT-related transcription factor ZEB1[14,15]. Claudin-low breast tumors are reported to be mostly estrogen receptor (ER)-negative, progesterone receptor (PR)-negative, and human epidermal growth factor receptor 2 (HER2)-negative (triple negative), and are associated with poor prognosis[12,16]. The prevalence of claudin-low breast cancer shows striking variability, ranging from 1.5% to 14% of tumors in breast cancer cohorts[5,7,8,12].

An algorithm (predictor) for identifying claudin-low tumors was described with the original characterization of the subtype[12]. Briefly, nine claudin-low cell lines were identified by hierarchical clustering of gene expression values of 1906 breast cancer intrinsic genes[17] in a cohort of 52 cell lines. Cell lines were used to build the claudin-low predictor, rather than bulk tumor samples, to minimize immune and stromal infiltration as confounding factors[12]. Two centroids were then defined: one for the cell lines with claudin-low gene expression features and one for all other breast cancer cell lines. Claudin-low tumors are identified by correlating a tumor's gene expression values to the two centroids and defining a tumor as claudin-low if it has stronger correlation to the claudin-low centroid than the other centroid. Importantly, the intrinsic subtypes (basal-like, HER2-enriched, luminal A, luminal B and normal-like) are first identified using the PAM50 predictor[17], and claudin-low subtyping is subsequently performed as an isolated second step[12]. In published studies, claudin-low is treated as a sixth intrinsic subtype, and the subtype assigned by PAM50 is therefore overwritten in claudin-low tumors[5,8,9,12]. As a consequence, claudin-low breast tumors have, thus far, been characterized as a single group, without regard for the distribution of the underlying intrinsic subtypes in the given set of claudin-low tumors[8,9,12,13].

In this study, we aim to elucidate the heterogeneity observed in claudin-low breast cancer. By stratifying claudin-low tumors according to intrinsic subtype, we show that the characteristics of claudin-low tumors reflect the intrinsic subtype to which they are initially assigned. Further, we explore an alternative method for identifying claudin-low tumors, and demonstrate that the nine-cell line claudin-low predictor[12] may be overly inclusive in classifying tumors with marked immune and stromal infiltration as claudin-low.

## Results

**Claudin-low breast tumors are delineated by intrinsic subtype.** We identified 87 claudin-low tumors (4.6%) in the METABRIC

cohort[4,5] using the nine-cell line claudin-low predictor[12,18] (Supplementary Data 1). By intrinsic subtype, the majority of these were classified either as basal-like (51.7%, $n = 45$), normal-like (32.2%, $n = 28$) or luminal A (LumA; 10.3%, $n = 9$) (Fig. 1a, Table 1). 14.6% and 15.3% of all basal-like and normal-like tumors, respectively, were identified as claudin-low. All three remaining subtypes were represented in the set of claudin-low tumors, but with a lower prevalence, representing 0.6 - 1.3% of tumors from each subtype. The distribution of intrinsic subtypes within the set of claudin-low tumors differed significantly from the distribution of intrinsic subtypes in non-claudin-low tumors ($P < 0.001$, $\chi^2$-test). Basal-like and normal-like tumors were significantly overrepresented in the set of claudin-low tumors, while the remaining intrinsic subtypes were significantly underrepresented ($P = 0.001$ for HER2-enriched, $P < 0.001$ for all other, Fisher's exact test). Only two HER2-enriched and three luminal B (LumB) tumors were classified as claudin-low. These two subtypes were not analyzed further due to low sample numbers. Claudin-low tumors broadly showed similar histology to non-claudin-low tumors (Supplementary Data 1), with 70% of tumors being classified as no special type (NST). One metaplastic tumor was found in the cohort, which was classified as claudin-low.

There were significant differences in the proportion of tumors expressing estrogen receptor when claudin-low tumors were stratified by intrinsic subtype (Fig. 1b; $P < 0.001$, $\chi^2$-test). 28.6%, 100% and 85.7% of basal-like, LumA, and normal-like claudin-low tumors, respectively, were ER-positive, closely reflecting the pattern seen in non-claudin low tumors (Fig. 1b). These findings indicate that the expression of ER in claudin-low tumors is reflected in their intrinsic subtype, and that characterizing claudin-low tumors as a triple negative subgroup of breast cancer[9,12] is an oversimplification.

Claudin-low tumors, as a whole, have previously been reported to have a low mutational burden and low level of genomic instability compared to the other subtypes[13,14]. Whole genome copy number data and sequence data from a panel of 173 cancer-associated genes were available for the METABRIC cohort[4,5]. When claudin-low tumors were stratified by intrinsic subtype, they consistently showed lower mutational burden and genomic instability compared to their non-claudin-low counterparts (Fig. 1c, d), with the exception of genomic instability in LumA tumors. There were, however, also significant differences in mutational burden ($P = 0.002$, Kruskal–Wallis test) and genomic instability ($P < 0.001$, Kruskal–Wallis test) between claudin-low tumors of the different intrinsic subtypes. Despite a degree of subtype specific variations, these findings point toward lower mutational rate and lower levels of genomic instability as bona fide claudin-low characteristics.

Curtis et al.[4] introduced breast cancer subtypes (IntClust) defined by patterns of CNA with *cis* correlation to gene expression. The genomically stable IntClust4 subtype showed overlap with claudin-low tumors[4,5,14]. In our analyses, 75% of all claudin-low tumors in the METABRIC cohort were classified as IntClust4. Stratified by intrinsic subtype, claudin-low tumors were consistently more likely to be classified as IntClust4 compared to non-claudin-low tumors of the same subtype (Fig. 1e). There were however significant variations in the proportion of claudin-low tumors classified as IntClust4 ($P < 0.001$, $\chi^2$-test), ranging from 60% of basal-like claudin-low tumors to 100% of normal-like claudin-low tumors. Further, IntClust4 tumors have been separated into ER-positive and ER-negative groups due to major differences in their biological and clinical characteristics, despite strong similarities in gene expression patterns and associated low levels of CNA[4,5,19]. Claudin-low tumors classified as IntClust4ER+ were predominantly LumA and normal-like, whereas claudin-low tumors classified as IntClust4ER− were predominantly basal-like (Supplementary Fig. 1a, b).

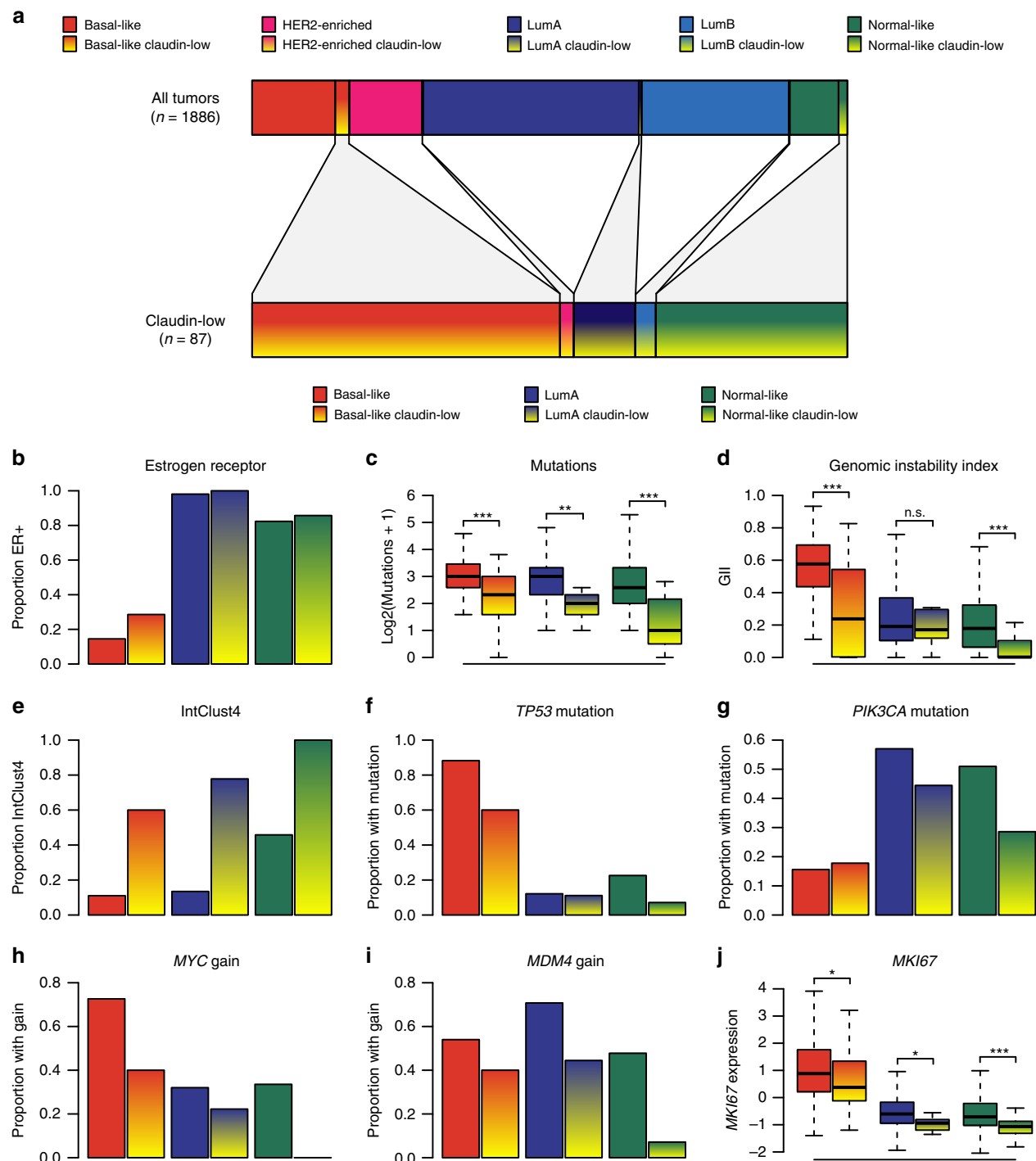

The high frequency of claudin-low tumors classified as IntClust4 supports the association between claudin-low gene expression characteristics and genomic stability. However, only 21% of all IntClust4 tumors in the METABRIC cohort were classified as claudin-low, and genomic instability index (GII) did not accurately predict correlation to the claudin-low centroid, as determined by the nine-cell line predictor[12] (Supplementary Fig. 2). Thus, while most claudin-low tumors were genomically stable, only a subset of genomically stable tumors were claudin-low.

No putative driver[20] mutations or CNAs were found at a significantly higher rate in claudin-low tumors, stratified by intrinsic subtype, than in non-claudin-low tumors of the same subtype (Fisher's exact test, Bonferroni corrected; Supplementary

Data 2). Rather, claudin-low tumors tended to exhibit patterns of mutation/CNA associated with their intrinsic subtype. Reflecting the lower levels of genomic instability and mutational burden, claudin-low tumors generally had lower incidences of potential driver aberrations compared to their non-claudin-low counterparts. To illustrate the relative frequencies of driver aberrations in claudin-low and non-claudin-low tumors, we selected four early genomic driver aberrations for further analysis: TP53 mutation, PIK3CA mutation, MYC gain (located on 8q24), and MDM4 gain (located on 1q32). Similar to the pattern observed for ER-positivity, the incidence of TP53 mutations in claudin-low tumors largely followed the incidence seen in the tumors' intrinsic subtype (Fig. 1f). The differences in TP53 mutation rates between

**Fig. 1 Claudin-low tumors are delineated by intrinsic subtype. a** Distribution of intrinsic subtypes in the METABRIC cohort for all tumors (top bar, $n =$ 1886) and for claudin-low tumors only (bottom bar, $n = 87$). **b** Estrogen receptor status. 58% of claudin-low tumors were ER-positive. ER prevalence differed between claudin-low tumors stratified by intrinsic subtype ($P < 0.001$, $\chi^2$-test). **c** Number of mutations in the panel of 173 sequenced genes. Claudin-low tumors showed lower mutational rates than non-claudin-low tumors of the same subtype (basal-like $P < 0.001$, LumA $P = 0.004$, normal-like $P < 0.001$, two-tailed Wilcoxon rank-sum test). **d** Genomic instability index (GII). Basal-like and normal-like claudin-low tumors showed lower levels of genomic instability than non-claudin-low tumors of the same subtype (basal-like $P < 0.001$, LumA $P = 0.83$, normal-like $P < 0.001$, two-tailed Wilcoxon rank-sum test). **e** IntClust4. 75% of claudin-low tumors were classified as IntClust4. IntClust4 classification differed between claudin-low tumors stratified by intrinsic subtype ($P < 0.001$, $\chi^2$-test). **f** TP53 mutation. 38% of claudin-low tumors carried TP53 mutations. Rate of TP53 mutation differed between claudin-low tumors stratified by intrinsic subtype ($P < 0.001$, $\chi^2$-test). **g** PIK3CA mutation. 24% of claudin-low tumors carried PIK3CA mutations. Differences between claudin-low tumors stratified by intrinsic subtype were not statistically significant ($P = 0.19$, $\chi^2$-test) **h** MYC gain. 26% of claudin-low tumors showed gain of MYC. Rate of MYC gain differed between claudin-low tumors stratified by intrinsic subtype ($P < 0.001$, $\chi^2$-test). **i** MDM4 gain. 30% of claudin-low tumors showed gain of MDM4. Rate of MDM4 gain differed between claudin-low tumors stratified by intrinsic subtype ($P = 0.006$, $\chi^2$-test). **j** MKI67 gene expression (log2). Claudin-low tumors consistently expressed lower levels of MKI67 compared to non-claudin-low counterparts (basal-like $P = 0.01$, LumA $P = 0.03$, normal-like $P < 0.001$, two-tailed Wilcoxon rank-sum test). **All** n.s. $P > 0.05$, *$P < 0.05$, **$P < 0.01$, ***$P < 0.001$. Sample sizes provided in Table 1. Boxplot elements: center line = median, box limits = upper and lower quartiles, whiskers = 1.5 × interquartile range. Source data are provided as a Source Data file.

**Table 1 Distribution of claudin-low tumors by intrinsic subtype in the METABRIC cohort.**

| Intrinsic subtype | Claudin-low ($n$) | Non- claudin-low ($n$) | Proportion claudin-low in subtype |
|---|---|---|---|
| Basal-like | 45 | 263 | 14.6% |
| HER2-enriched | 2 | 231 | 0.9% |
| LumA | 9 | 684 | 1.3% |
| LumB | 3 | 466 | 0.6% |
| Normal-like | 28 | 155 | 15.3% |

Source data are provided as a Source Data file.

claudin-low tumors stratified by intrinsic subtype were statistically significant ($P < 0.001$, $\chi^2$-test). There were similar trends for the other three aberrations analyzed (Fig. 1g–i). Claudin-low tumors stratified by intrinsic subtype showed significantly different rates of MYC and MDM4 gain ($P < 0.001$ and $P = 0.006$, $\chi^2$-test), but not PIK3CA mutation ($P = 0.19$, $\chi^2$-test).

Claudin-low tumors have previously been characterized as slower cycling, with proliferation levels lower than in basal-like tumors, but higher than in LumA and normal-like tumors[8,12]. Ki-67, encoded by the MKI67 gene, is a commonly used proliferation marker. When claudin-low tumors were stratified by intrinsic subtype, there were significantly different levels of MKI67 expression between subtypes (Fig. 1j; $P < 0.001$, Kruskal–Wallis test), with basal-like claudin-low tumors showing significantly higher levels of MKI67 expression than LumA claudin-low tumors and normal-like claudin-low tumors ($P < 0.001$ for both, Wilcoxon rank-sum test). Claudin-low tumors did, however, also show significantly lower levels of MKI67 expression than non-claudin-low counterparts in all intrinsic subtypes (Fig. 1j; $P = 0.01$, 0.03 and $< 0.001$ claudin-low compared to non-claudin-low in basal-like, LumA, and normal-like tumors, respectively, Wilcoxon rank-sum test). Thus, MKI67 gene expression levels indicate that claudin-low tumors reflect the proliferation levels of their intrinsic subtype but are also slower cycling than non-claudin-low counterparts.

Claudin-low tumors have previously been associated with poor prognosis[8,12]. This characterization was accurate when claudin-low tumors were viewed as a single group (Supplementary Fig. 1c). However, when the survival of patients with claudin-low tumors was stratified by intrinsic subtype, the survival patterns generally observed in non-claudin-low breast cancer[2] re-emerged (Fig. 2a). Further, there were no significant differences in survival

between patients with claudin-low and non-claudin-low tumors within each intrinsic subtype (Fig. 2b–d). Thus, we did not find evidence indicating that claudin-low status affects survival in breast cancer patients.

Claudin-low tumors have been reported to mostly occur in younger patients, with age at diagnosis slightly higher than in basal-like tumors, but lower than in the remaining subtypes[8,9]. When claudin-low tumors were stratified by intrinsic subtype, there were, however, significant differences in the average age at diagnosis ($P = 0.01$, Kruskal–Wallis test; Supplementary Fig. 1d), with basal-like claudin-low tumors diagnosed at a significantly lower age than LumA claudin-low and normal-like claudin-low tumors ($P = 0.03$ and 0.01 respectively, Wilcoxon rank-sum test). Claudin-low and non-claudin-low tumors of the same intrinsic subtype showed similar age at diagnosis (basal-like $P = 0.67$, LumA $P = 0.53$, normal-like $P = 0.052$, two-tailed Wilcoxon rank-sum test).

**A condensed gene list refines claudin-low classification.** Claudin-low tumors have been shown to exhibit high degrees of immune and stromal infiltration[9,12]. Also when stratified by intrinsic subtype, claudin-low tumors in the METABRIC cohort consistently had higher infiltration of immune and stromal cells compared to non-claudin-low tumors (as determined by ESTI-MATE, a gene expression-based tool for inferring normal-cell infiltration in tumors[21]) (Supplementary Fig. 3a, b). The nine-cell line claudin-low predictor uses 807 genes, and Prat et al. acknowledge that it may inappropriately identify some tumors as claudin-low solely due to stromal infiltration[12]. This statement is supported by a strong correlation between a tumor's inferred[21] stromal infiltration and closeness to the nine-cell line claudin-low centroid ($R^2 = 0.76$, linear regression; Supplementary Fig. 3c, d). A similar, but weaker trend was also observed for inferred[21] immune cell infiltration ($R^2 = 0.27$, linear regression; Supplementary Fig. 3e, f). We therefore considered whether a more targeted gene list could be used for claudin-low classification, in order to more accurately isolate features intrinsic to claudin-low tumors.

We created a condensed claudin-low gene list (Supplementary Table 1), consisting of 19 genes representing only the pathognomonic gene expression characteristics of claudin-low tumors: Low expression of cell–cell adhesion genes, high expression of EMT genes, and gene expression patterns typical of stem cell-like/less differentiated cells[3,8,9,12,14]. In the METABRIC cohort, hierarchical clustering of gene expression values, using the condensed gene list, revealed a tumor cluster with gene expression characteristics in line with those previously described in

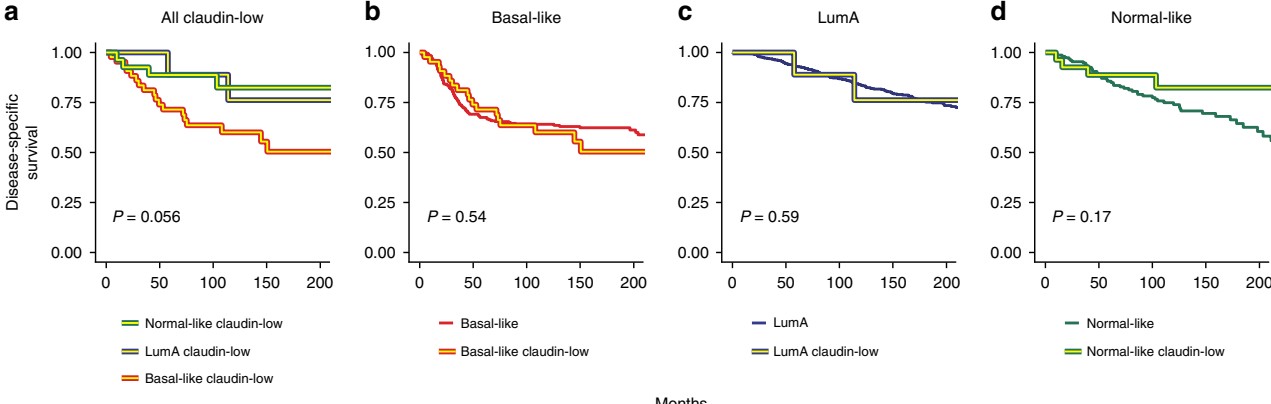

**Fig. 2 No evidence of claudin-low status as an indicator of poor prognosis in the METABRIC cohort. a** Disease-specific survival in basal-like claudin-low, LumA claudin-low, and normal-like claudin-low tumors in the METABRIC cohort. Survival trends recapitulated the patterns seen in non-claudin-low tumors. $P = 0.056$ when testing for difference between claudin-low tumors stratified by intrinsic subtype. **b–d** Disease-specific survival in claudin-low and non-claudin-low basal-like (**b**), LumA (**c**) and normal-like (**d**) tumors. Significant differences between claudin-low and non-claudin-low tumors were not found (basal-like $P = 0.54$, LumA $P = 0.59$, normal-like $P = 0.17$). **All** Two-tailed log-rank test used for significance testing. Disease-specific deaths and sample sizes: basal-like claudin-low $n = 19$ of 45, basal-like non-claudin-low $n = 98$ of 263, LumA claudin-low $n = 3$ of 9, LumA non-claudin-low $n = 144$ of 684, normal-like claudin-low $n = 5$ of 28, normal-like non-claudin-low $n = 50$ of 155. Source data are provided as a Source Data file.

claudin-low tumors (Fig. 3; $P = 0.006$, SigClust[22]). We refer to tumors in this cluster as core claudin-low (CoreCL), while claudin-low tumors (as defined by the nine-cell line predictor) outside the CoreCL cluster are referred to as other claudin-low (OtherCL). Individual inspection of gene expression values showed that OtherCL tumors displayed certain claudin-low characteristics, albeit to a lesser degree than CoreCL tumors (Supplementary Fig. 4).

The CoreCL cluster consisted of 79 tumors (4.2% of tumors in the cohort), of which 57 (72.2%) were identified as claudin-low by the nine-cell line predictor (Supplementary Data 1). While several intrinsic subtypes were prominently represented in the group of CoreCL tumors, the OtherCL ($n = 30$) tumors were predominantly basal-like ($n = 23$; Fig. 4a). Thus, our method for identifying claudin-low tumors primarily differed from the nine-cell line predictor by filtering out a set of basal-like tumors with high levels of stromal and immune infiltration (Supplementary Fig. 5a, b), but without pathognomonic claudin-low gene expression characteristics (Supplementary Fig. 6).

There were marked contrasts between the characteristics of basal-like CoreCL tumors ($n = 25$), basal-like OtherCL-tumors ($n = 23$), and non-claudin-low basal-like tumors ($n = 260$). Basal-like CoreCL tumors carried significantly fewer mutations than basal-like OtherCL tumors and non-claudin-low basal-like tumors (Fig. 4b; $P = 0.015$ & $P < 0.001$ respectively, Wilcoxon rank-sum test). Basal-like CoreCL tumors also displayed significantly lower levels of genomic instability than basal-like OtherCL tumors and non-claudin-low basal-like tumors (Fig. 4c; $P < 0.001$ for both, Wilcoxon rank-sum test). There were no significant differences in GII between basal-like OtherCL tumors and non-claudin-low basal-like tumors (Fig. 4c, $P = 0.082$, Wilcoxon rank-sum test). There was also a greater proportion of basal-like CoreCL tumors in IntClust4, than basal-like OtherCL and non-claudin-low basal-like tumors (Fig. 4d, Supplementary Fig. 5c, d). In total, 80% of basal-like CoreCL tumors were classified as IntClust4, in contrast to 43% of basal-like OtherCL tumors and 10% of basal-like non-claudin-low tumors. There were also lower rates of *TP53* mutation, *MYC* gain and *MDM4* gain, in basal-like CoreCL tumors compared to basal-like OtherCL and basal-like non-claudin-low tumors, reflecting the lower mutational burden and GII (Supplementary Fig. 5e, g). This trend was, however, not evident for *PIK3CA* (Supplementary

Fig. 5h). Basal-like CoreCL tumors expressed significantly lower levels of *MKI67* than basal-like OtherCL and basal-like non-claudin-low tumors (Fig. 4e; $P < 0.001$ for both, Wilcoxon rank-sum test). There were no significant differences in *MKI67* expression between basal-like OtherCL and basal-like non-claudin-low tumors ($P = 0.63$, Wilcoxon rank-sum test). In sum, the characteristics of basal-like OtherCL tumors show weaker concordance with the characteristics of claudin-low tumors, compared to basal-like CoreCL tumors. It is therefore likely that OtherCL tumors are classified as claudin-low by the nine-cell line predictor due to their stromal infiltration (Supplementary Figs. 3c, 5b), and that the classification of these tumors as claudin-low may be dubious.

Despite differences in genomic and transcriptomic features, as well as in immune and stromal infiltration, there were no significant differences in survival between basal-like CoreCL, basal-like OtherCL and non-claudin-low basal-like tumors (Fig. 4f). These findings reinforce our observations indicating that claudin-low status is not a major determinant of survival in breast cancer patients.

There were few OtherCL samples not classified as basal-like ($n = 1$, 3, and 3 for LumA, LumB and normal-like tumors, respectively; Fig. 4a). The characteristics of normal-like CoreCL ($n = 39$) and LumA CoreCL ($n = 13$) tumors were similar to the characteristics of normal-like claudin-low and LumA claudin-low tumors identified by the nine-cell line predictor (Supplementary Figs. 7, 8). These findings indicate that the nine-cell line predictor's promiscuous classification of stromally infiltrated tumors as claudin-low may mostly be of concern in basal-like tumors.

**Validation cohorts reinforce key claudin-low characteristics.** To validate our findings, we queried the Oslo2 cohort[23], for which gene expression data and ER/HER2 status were available. There were 29 claudin-low tumors, as defined by the nine-cell line predictor, in the cohort (7.6%), of which most were classified as basal-like, LumA or normal-like ($n = 7$, 5 and 11, respectively; Supplementary Fig. 9a, Supplementary Data 3). When clustering using the condensed claudin-low gene list, there was a cluster with claudin-low gene expression characteristics and high levels of immune and stromal cell infiltration (Supplementary Fig. 9b;

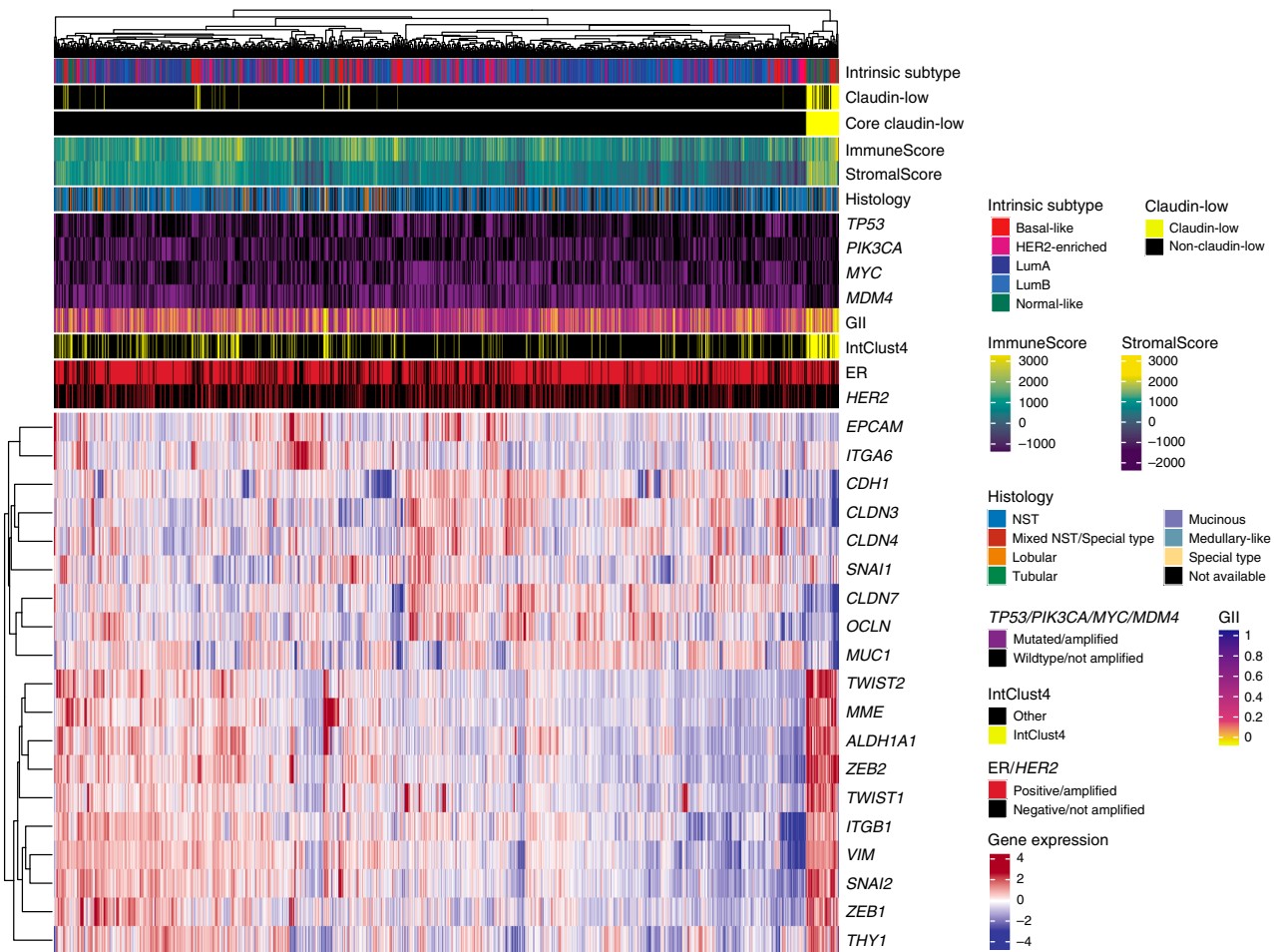

**Fig. 3 A condensed claudin-low gene list identifies a set of core claudin-low tumors.** Heatmap of gene expression values (log2) for a condensed claudin-low gene list in the METABRIC cohort (n = 1886 biologically independent samples). A cluster, marked Core claudin-low (n = 79), emerged with transcriptomic and genomic claudin-low characteristics (P = 0.006, SigClust[22]). Source data are provided as a Source Data file.

P < 0.001, SigClust[22]). 28 tumors in the cohort (7.3%) were located in the core claudin-low cluster (Supplementary Fig. 9c), of which 16 (57%) were identified as claudin-low by the nine-cell line predictor. Seven basal-like tumors were classified as claudin-low by the nine-cell line predictor; two of these were CoreCL, both of which were IntClust4, and the remaining five were OtherCL, none of which were IntClust4. Using IntClust4 as a surrogate marker for low levels of genomic instability[4,19,24], these findings emphasize that the nine-cell line predictor may be overly inclusive in identifying basal-like tumors as claudin-low. The OtherCL tumors in the Oslo2 cohort were, however, more diverse than in the METABRIC cohort, with 7 of 12 OtherCL tumors being non-basal-like (n = 1, 4 and 2 for HER2-enriched, LumA, and LumB, respectively). In total, 89% of CoreCL tumors in the Oslo2 cohort were classified as IntClust4, compared to 38% of OtherCL tumors and 20% of non-claudin-low tumors. Thus, the characteristics of claudin-low tumors in the Oslo2 cohort were mostly consistent with those observed in the METABRIC cohort.

Finally, we explored the TCGA breast cancer cohort[7,25]. 32 of 1082 tumors (3.0%) were classified as claudin-low by the nine cell-line predictor (Supplementary Data 4); however, no core claudin-low cluster emerged (Supplementary Fig. 10). As previously noted, non-tumor cell infiltration is a central characteristic of claudin-low tumors. An inclusion criterion in the TCGA protocol is a tumor cellularity over 60%[7]. The METABRIC cohort was originally divided into a discovery cohort with a cellularity cut-off of 40%, which had a claudin-low prevalence of 3.6%, and a validation cohort with no cellularity cut-off[4], which had a claudin-low prevalence of 5.6%. There was no cut-off for cellularity in the Oslo2 cohort[23], which had a claudin-low prevalence of 7.6%. Thus, there may be an association between cellularity cut-off in a cohort and claudin-low prevalence (Fig. 5). This strengthens the observation of non-tumor cell infiltration as a fundamental claudin-low characteristic and may explain the absence of a core-claudin-low cluster in the TCGA-BRCA cohort.

Claudin-low tumors in the TCGA breast cancer cohort mostly showed histological features in line with those of non-claudin-low tumors (Supplementary Data 4). There were eight metaplastic tumors in the cohort, of which six were classified as claudin-low, confirming that most metaplastic tumors are claudin-low[26].

## Discussion

Here, we have re-evaluated the characteristics of claudin-low breast tumors, from the perspective of claudin-low as a phenotype that may permeate the intrinsic subtypes. Through analyses of genomic, transcriptomic and clinical data, we have shown that the characteristics of claudin-low tumors reflect their intrinsic subtype. Characteristics that are associated with claudin-low status include marked immune and stromal cell infiltration, low levels of genomic instability and mutational burden, and reduced proliferation levels. Finally, we explored an alternative method for identifying claudin-low tumors, and thereby showed that a subset of tumors with pronounced immune and stromal infiltration may

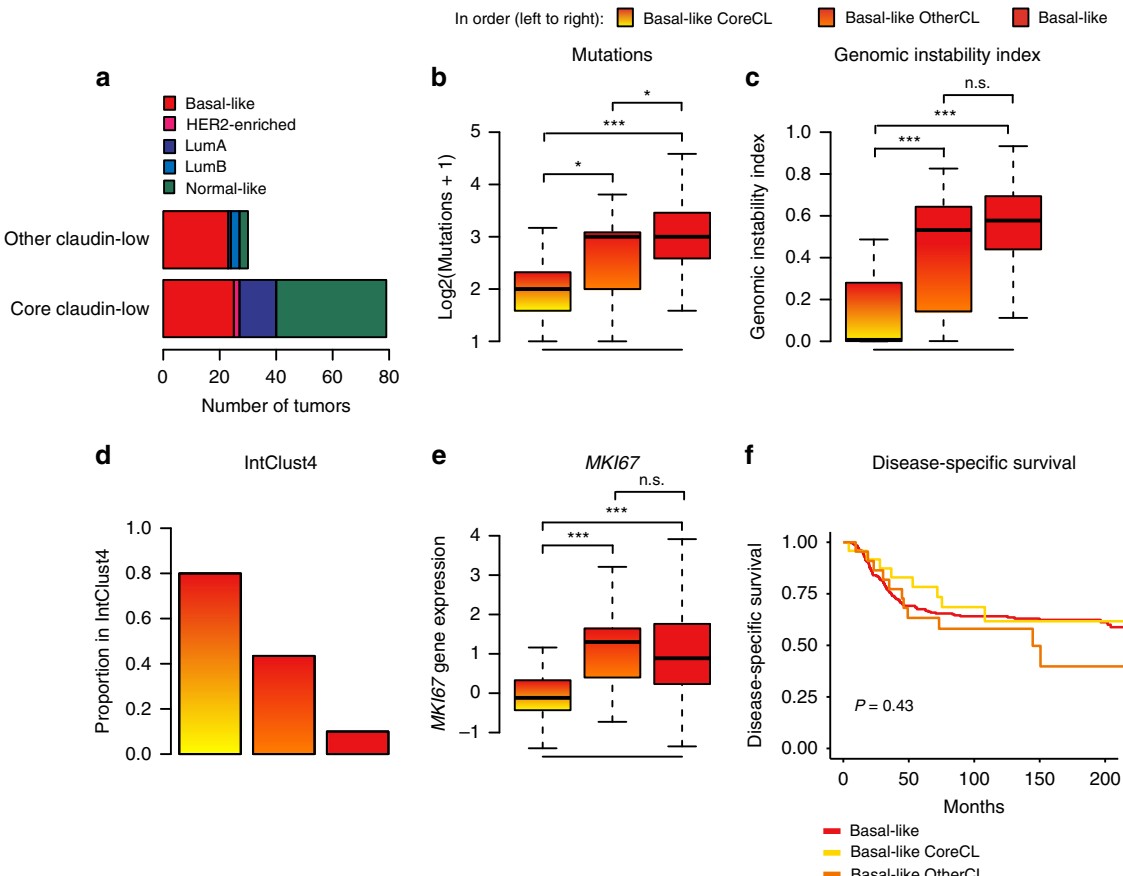

**Fig. 4 Basal-like OtherCL tumors may be inappropriately classified as claudin-low. a** Distribution of subtypes in CoreCL and OtherCL tumors in the METABRIC cohort. Hierarchical clustering with the condensed claudin-low gene list filtered out a subset of basal-like claudin-low tumors (as defined by the nine-cell line predictor) with weak claudin-low characteristics. **b** Number of mutated genes in the panel of 173 sequenced genes. Basal-like CoreCL tumors carried significantly fewer mutations than basal-like OtherCL tumors and basal-like non-claudin-low tumors ($P = 0.02$ and $P < 0.001$ respectively, two-tailed Wilcoxon rank-sum test). **c** Distribution of genomic instability index. Basal-like CoreCL tumors showed significantly lower levels of genomic instability than basal-like OtherCL tumors and non-claudin-low basal-like tumors ($P < 0.001$ for both, two-tailed Wilcoxon rank-sum test). **d** Proportion of tumors in IntClust4. 80% of basal-like CoreCL tumors were classified as IntClust4. **e** *MKI67* gene expression (log2). Basal-like CoreCL tumors expressed significantly lower levels of *MKI67* than basal-like OtherCL and non-claudin-low tumors ($P < 0.001$ for both, two-tailed Wilcoxon rank-sum test). **f** Disease-specific survival in basal-like CoreCL, basal-like OtherCL and non-claudin-low basal-like tumors. Disease-specific survival in basal-like breast tumors did not significantly differ when stratified by claudin-low status (two-tailed log-rank test). Disease-specific deaths: basal-like CoreCL $n = 8$ of 25, basal-like OtherCL $n = 12$ of 23, basal-like non-claudin-low $n = 97$ of 260. **All** n.s. $P > 0.05$, *$P < 0.05$, **$P < 0.01$, ***$P < 0.001$. Boxplot elements: center line = median, box limits = upper and lower quartiles, whiskers = $1.5 \times$ interquartile range. Basal-like CoreCL $n = 25$, basal-like OtherCL $n = 23$, basal-like non-claudin-low $n = 260$, HER2-enriched CoreCL $n = 2$, LumA CoreCL $n = 13$, LumA OtherCL $n = 1$, LumB OtherCL $n = 3$, normal-like CoreCL $n = 39$, normal-like OtherCL $n = 3$ biologically independent samples. Source data are provided as a Source Data file.

be inappropriately classified as claudin-low by the established claudin-low predictor[12].

We stratified claudin-low tumors by intrinsic subtype and found differences between claudin-low tumors of different intrinsic subtypes in almost all aspects analyzed. Perhaps most surprisingly, we found no evidence indicating that claudin-low status affects disease-specific survival, contrasting with previous reports of claudin-low as a poor prognosis subtype[8,12]. These findings imply that a large subset of previously reported characteristics of claudin-low tumors are not bona fide claudin-low characteristics but are rather an average of the characteristics of several intrinsic subtypes. Thus, the established practice of analyzing claudin-low tumors as a single entity, without taking intrinsic subtype into consideration, may obscure the features that are attributable to claudin-low status.

Claudin-low breast cancer has previously been considered a single disease entity, analogous to the intrinsic breast cancer subtypes[8,9,12,13] (Fig. 6a). Our findings, however, imply that

breast tumors are not claudin-low instead of the intrinsic subtype to which they are assigned by the PAM50 predictor, rather that they can carry a claudin-low phenotype in addition to their intrinsic subtype (Fig. 6b). According to this interpretation, claudin-low is a measure of a set of biological features which is distinct from the set of biological features measured by the intrinsic subtypes.

We explored a method of identifying claudin-low tumors using a condensed gene list. The claudin-low tumors identified using this method (CoreCL) showed more consistent traits than the claudin-low tumors identified by the nine-cell line predictor. OtherCL tumors can be interpreted to not be genuine claudin-low tumors. OtherCL tumors did, however, display some genomic and transcriptomic traits which were consistent with the claudin-low phenotype, though to a lesser degree than CoreCL tumors. A compelling interpretation may instead be that claudin-low is a continuum (degree of "claudin-lowness", Fig. 6c), rather than a binary feature (claudin-low vs. non-claudin-low, Fig. 6b).

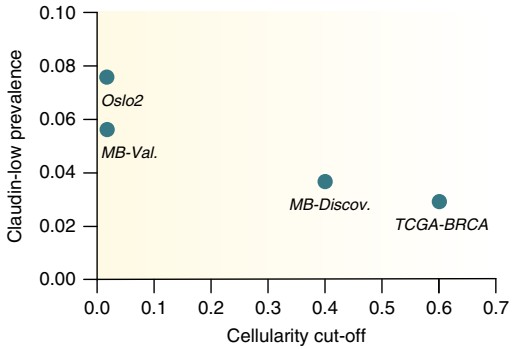

**Fig. 5 Cut-offs for tumor cellularity may affect the prevalence of claudin-low tumors in breast cancer cohorts.** Relationship between cut-offs for tumor cellularity in a cohort and the prevalence of claudin-low tumors (as identified by the nine-cell line predictor). Cohorts which excluded tumors due to low cellularity had a lower claudin-low prevalence than cohorts without a cellularity exclusion criterion. MB-Discov.: METABRIC discovery cohort, MB-Val.: METABRIC validation cohort. MB-Discov. $n = 957$, MB-Val. $n = 929$, Oslo2 $n = 381$, TCGA-BRCA $n = 1082$ biologically independent samples. Source data are provided as a Source Data file.

According to this interpretation, breast tumors might exist along a spectrum of claudin-lowness, in which they lie somewhere between: (1) non-claudin-low, fully concordant with an intrinsic subtype, (2) moderately claudin-low with marked imprint of an intrinsic subtype (exemplified by the average claudin-low tumor identified by the nine-cell line predictor), (3) extensively claudin-low, with limited imprint of an intrinsic subtype (exemplified by the average CoreCL tumor), or (4) purely claudin-low, with no imprint of intrinsic subtype (perhaps exemplified by special histological subtypes[26,27]). This model would be consistent with recent descriptions of partial EMT phenotypes in cancer[28,29] and cellular pliancy as an etiological explanation for the claudin-low phenotype[14,15].

Claudin-low tumors had high levels of non-tumor cell infiltration, and there was a lower prevalence of claudin-low tumors in the cohorts with a cut-off for tumor cellularity. It is also known that EMT-like gene expression features in tumors are similar to the gene expression characteristics of stromal tissue[29], and a subset of normal breast tissue samples show marked similarities to claudin-low-like gene expression patterns[30,31]. In the context of these observations, it is pertinent to ask: How much of the claudin-low phenotype is a result of stromal infiltration, and could the claudin-low phenotype simply be an artifact of stromal infiltration? If the claudin-low phenotype were only a sampling artifact, irrelevant to a tumor's biology, one would expect claudin-low tumors to be similarly distributed among the intrinsic subtypes. Claudin-low tumors were, however, overrepresented in basal-like and normal-like tumors, and underrepresented in the remaining intrinsic subtypes. Further, if claudin-lowness were only mediated by stromal infiltration, it should be possible to accurately identify claudin-low tumors solely on the basis of stromal infiltration. However, while almost all claudin-low tumors had high levels of stromal infiltration, only a minority of tumors with high levels of stromal infiltration were in fact classified as claudin-low (Supplementary Data 1, 3, and 4). Finally, numerous studies have identified features in claudin-low tumors (human, murine and cell-line), which are directly attributable to claudin-low tumor cells[6,9,12,14,32]. Therefore, non-tumor cell infiltration is undoubtedly an important feature of the claudin-low tumor microenvironment[33–36], and may even be the feature that induces EMT in claudin-low tumor initiating cells[15].

However, the characteristics observed in claudin-low tumors cannot solely be attributed to non-tumor-cell infiltration.

While we did not find evidence that claudin-low status affects survival, certain claudin-low characteristics may nonetheless be clinically relevant and/or actionable. For example, claudin-low tumors show high levels of immune cell infiltration[8,12], express high levels of *PD-L1*[13], are immunosuppressed by T-regulatory cells[33], and carry low mutational burden;[13,14] these factors may all be relevant for the efficacy of immunotherapeutics in claudin-low tumors. The EMT phenotype in claudin-low tumors may in itself be a therapeutic target, and may also have implications for chemoresistance[37]. Due to the major influence of non-tumor-cell infiltration, it is likely that immunocompetent animal models will be of particular importance for functionally evaluating how these features can be therapeutically targeted[3,13,38,39].

Several limitations of this study should be noted. Despite analyzing over three thousand breast tumors, we identified relatively few claudin-low tumors. The findings presented in this article must therefore be interpreted with a degree of caution. While the Kaplan–Meier curves for the METABRIC cohort (Fig. 2, Supplementary Fig. 8) show clear resemblance to those observed in non-claudin-low tumors[2], the claudin-low cohort was not powered to detect statistically significant differences between groups. Further, it is difficult to ascertain the extent to which the observations from bulk tumor samples represent the characteristics of tumor cells or non-tumor cells[29]. It is therefore likely that single-cell transcriptomic analyses will be required in order to effectively disentangle the features of tumor cells and infiltrating immune and stromal cells. Finally, it must be highlighted that we deliberately chose a biased approach to building the condensed claudin-low gene list. This choice was motivated by our findings (Supplementary Fig. 3) and informed by contemporary studies of claudin-low tumors[6,9,14,15,32]. We therefore stress that there is no gold standard for identifying claudin-low tumors, and that the method presented here may lack external generalizability. Additional approaches to refining claudin-low classification, which could be used in conjunction with the nine-cell line predictor or the method presented here, might include: Immunohistochemical staining of EMT-related protein markers, implementing a cut-off for maximum permitted GII, or checking for overlap with IntClust4 status.

In summary, we have comprehensively analyzed claudin-low breast tumors, and through these analyses substantiated a re-definition of claudin-low as a breast cancer phenotype. Our findings explain the large degree of heterogeneity observed in claudin-low breast tumors, thereby enabling more accurate and nuanced investigations into this poorly understood form of cancer.

## Methods
**Cohorts**. The METABRIC[4,5], Oslo2[23], and TCGA-BRCA[7,25] cohorts were analyzed in this study. Processed data from the METABRIC cohort were downloaded from cBioportal;[40,41] queried data include hormone receptor status, IntClust subtype, disease-specific survival, mutation status in a panel of 173 sequenced genes[5], gene-centric copy number status, and normalized gene expression values. Intrinsic subtypes (identified using the PAM50 predictor[17]) for the METABRIC cohort were retrieved from supplementary files in Curtis et al.[4]. Copy number segments and tumor ploidy were retrieved from the repository associated with Pereira et al.[5]. There were 1886 tumors in the METABRIC cohort with aforementioned data available. Centrally reviewed histological classifications were kindly made available by Dr. Elena Provenzano[42]. Histological classification was available for 1575 tumors in the dataset.

For the Oslo2 cohort, normalized gene expression values, intrinsic subtypes (identified using PAM50) and hormone receptor status were downloaded from the Gene Expression Omnibus (GEO), accession GSE80999. All 381 samples from the cohort were included in the analyses. Analyses were carried out using GEOquery[43] and Biobase[44]. Copy number data were only available for seven claudin-low tumors and was therefore not used in the analyses.

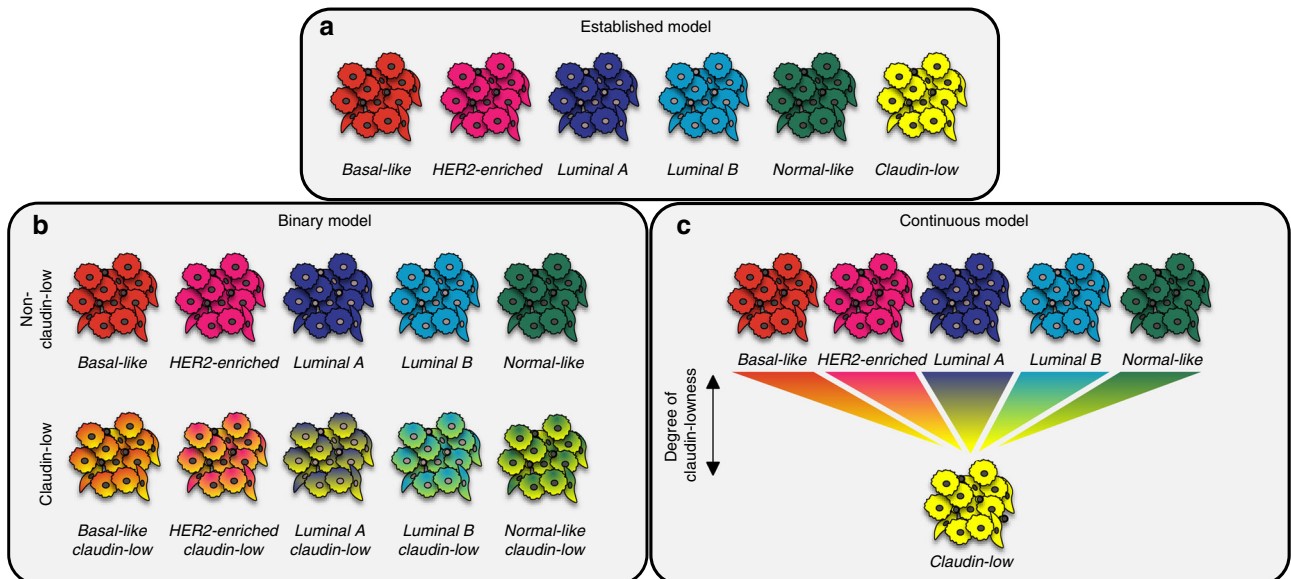

**Fig. 6 Re-definition of claudin-low as a breast cancer phenotype. a** In the established model, claudin-low is interpreted as a sixth subtype, analogous to the intrinsic subtypes. **b** When stratified by intrinsic subtype, claudin-low tumors, however, show characteristics associated with their intrinsic subtype. This implies that tumors are not claudin-low instead of their intrinsic subtype, rather that tumors can carry a claudin-low phenotype in addition to their intrinsic subtype. In the binary model, a tumor is either classified as claudin-low, or non-claudin-low. **c** The comparative analysis of CoreCL tumors and claudin-low tumors identified by the nine-cell line predictor, indicates that claudin-low may in fact be a continuous feature. Thus, individual tumors may show varying degrees of claudin-lowness, rather than simply being claudin-low or non-claudin-low. In this model, CoreCL tumors, on average, have a higher degree of claudin-lowness than claudin-low tumors identified by the nine-cell line predictor. The continuous model opens for the possibility of pure claudin-low tumors, uncoupled from the intrinsic subtypes.

Normalized gene expression values, intrinsic subtype (identified using PAM50) and histological classification from tumors in the TCGA-BRCA cohort were downloaded from cBioportal[7,25,40,41]. All 1082 tumors from the TCGA-BRCA cohort were analyzed.

**Transcriptomic analyses.** The generation and pre-processing of gene expression data are described in the cohorts' respective publications[4,5,7,23,25]. Gene expression values were mean centered and scaled (z-score). In the Oslo2 cohort, genes represented by multiple probes were reduced to a single gene expression value by finding the mean of all probes representing the given gene.

Claudin-low tumors were identified using the implementation of the nine-cell line claudin-low predictor[12] in the Genefu[18] package for R[45]. Euclidean distance was used as the distance metric for claudin-low classification. IntClust subtypes in the Oslo2 and TCGA-BRCA cohorts were determined using a gene-expression-based IntClust-classifier[24] implemented in Genefu[18]. Immune and stromal infiltration was inferred from gene expression data using *ImmuneScore* and *StromalScore* derived by ESTIMATE[21].

We observed that the nine-cell line claudin-low predictor was heavily influenced by the effect of non-tumor cell infiltration (Supplementary Fig. 3). This can be related to the marked stromal and immune infiltration in claudin-low tumors[12], and to the partial overlap in gene expression features between stromal tissue and tumors with an EMT phenotype[29–31]. Given these challenges which arose from the unbiased approach used by Prat et al.[12], and the progress made in the understanding of claudin-low tumors[6,14,15,32], we chose to explore a biased approach to identifying claudin-low tumors. The reduced gene set used to identify core claudin-low tumors (Supplementary Table 1) was manually selected on the basis of published characterizations of claudin-low gene expression features and advances in understanding the etiological basis of claudin-low tumors[3,6,8,9,12,14,15,32]. We reasoned that the genes should capture the characteristics unique to claudin-low tumors: Low expression of cell–cell adhesion genes, high expression of EMT genes, and stem-cell like/undifferentiated gene expression pattern. Further, we reasoned that the gene list should not include characteristics that are not unique to claudin-low tumors, such as a low expression of luminal epithelium-related genes. Inclusion of such genes would risk recapitulating the intrinsic subtypes. Hierarchical clustering using the reduced gene list was performed by complete linkage with Euclidean distance as the distance metric. Clustering and visualization were performed using the ComplexHeatmap package[46] for R. The significance of the core claudin-low cluster was evaluated using SigClust[22].

**Genomic analyses.** GII was derived by dividing the number of copy number aberrant nucleotides by the total number of nucleotides in the genome. GII was ploidy-corrected by defining a segment as copy number aberrant if the copy number state deviated from the nearest integer value for ploidy. All GII values were ploidy-corrected.

Individually analyzed genomic aberrations were chosen according to the following criteria: (1) known function as early driver events;[20,47] (2) among the most frequently observed aberrations in breast cancer;[4,5] (3) significantly different incidence between intrinsic subtypes ($\chi^2$-test $P < 0.05$);[4,5] (4) non-overlap with other selected events (i.e. only one CNA located on 8q24). *TP53* mutation, *PIK3CA* mutation, *MYC* amplification (8q24), and *MDM4* amplification (1q32) were selected for further analysis on the basis of these criteria.

**Survival analyses.** Survival analyses were performed using the Survival package[48] for R, and Kaplan–Meier plots were generated using the Survminer package.

**Statistical analyses.** All significance tests (where applicable) were two-tailed. For continuous variables, Wilcoxon rank-sum test and Kruskal–Wallis test were used to test for differences between two or more than two groups, respectively. For categorical variables, Fisher's exact test and $\chi^2$-test were used to test for differences between two or more than two groups, respectively. Significance in survival analyses was determined by log-rank tests. Adjustments were made for multiple hypothesis testing in the analyses detailed in Supplementary Data 2 (Bonferroni-correction); no other corrections were made for multiple hypothesis testing. Whiskers in box-and-whisker plots were generated using the Tukey method; individual data points were not plotted, as the imbalance in sample numbers between groups (Table 1) tended to obscure overall trends.

**Reporting summary.** Further information on research design is available in the Nature Research Reporting Summary linked to this article.

## Data availability

The data used in this study are available through cBioportal[40,41] (METABRIC[4,5], TCGA[7,25], GSE80999[23], supplementary tables 2 and 3 in Curtis et al.[4] and the repository associated with Pereira et al.[5]. Histological classification of the METABRIC cohort may be available upon request to Mukherjee et al.[42] Detailed instructions for gathering data can be found in the repository associated with this study. The source data underlying each figure are provided as a Source Data file. All other data are available within the

Article, Supplementary Information files or available from the author upon reasonable request.

## Code availability

All code used in the described analyses is available at https://github.com/clfougner/ClaudinLow.

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

## Acknowledgements

The authors thank Aleix Prat and Ole Christian Lingjærde for insightful discussions and critical reading of the manuscript. We are grateful to Elena Provenzano for providing us with centrally reviewed histological classifications of tumors in the METABRIC cohort, Hege G. Russnes for histopathological support, and the Oslo Breast Cancer Research Consortium (OSBREAC) for access to data from the Oslo2 cohort. C.F., H.B., and J.H.N. are supported by grants from the Norwegian Research Council (163027) and South-Eastern Norway Regional Health Authority (2012056) to T.S.

## Author contributions

C.F. conceptualized and designed the study, and performed all analyses. C.F., H.B., J.H.N., and T.S. interpreted the results. J.H.N. and T.S. provided supervision. T.S. acquired funding. C.F. wrote the original manuscript draft. C.F., H.B., J.H.N., and T.S. reviewed and edited the manuscript.

## Competing interests

The authors declare no competing interests.
