## [Peer Review File · Nature Communications]

Reviewers' comments:

Reviewer #1 (Remarks to the Author):

In this manuscript the authors investigate genomic, transcriptomic, and clinical features of claudin-low breast tumors. The claudin-low breast cancer subtype that was initially identified by clustering human tumors and murine breast cancer (BRCA) models and then a classifier built using human cell lines to minimize immune and stromal infiltration. It is characterized by low expression of cell-cell adhesion genes, high expression of epithelial-mesenchymal transition (EMT) genes, and stem cell-like/less differentiated gene expression patterns.

The authors have identified 87 claudin-low tumors from the METABRIC cohort and performed analysis of ER, mutation burden, genomic instability, MKI67 and specific oncogenic mutations within the intrinsic subtypes. Within each of the intrinsic subtypes, claudin-low tumors were distinguished by low genomic instability, mutational burden and proliferation levels, and high levels of immune and stromal cell infiltration. The authors show no evidence that claudin-low cancers are an indicator of poor prognosis when binned by intrinsic subtype, as previously described by others when pooled together. They provide compelling evidence that claudin-low is not simply a subtype as previously portrayed, but is a complex additional phenotype which span various intrinsic subtypes. They have developed an alternative method for identifying claudin-low tumors using a biased gene set and uncovered potential weaknesses in the established claudin-low classifier.

These findings suggest heterogeneity in claudin-low breast tumors, and substantiate a re-definition of claudin-low as a cancer phenotype. They also provide a reasonable argument that tumor cellularity contributes to the varying degrees of claudin-low-tumors (1.5-14%) between differing datasets. Given the lower mutation, decreased genomic instability, high percentage of stroma (by ESTIMATE) and the varying amounts by tumor cellularity, these data also suggest that perhaps a portion of claudin-low tumors are likely tumors from any intrinsic subtype with a high proportion of normal stroma. The authors should consider additional studies of H&E sections along with histopathological characteristics of tumors such as metaplastic features in defining the subtype. The potential confounding issue of normal stroma could be demonstrated with additional evaluation of patient matched tumor and adjacent normal tissue.

Major Concerns

1. The fact that a large proportion of normal-like subtype are claudin-low, combined with the fact the claudin low occur across all subtypes, with lower genomic instability, copy number and mutational burden, and the number of claudin-low tumors is dependent on tumor cellularity strongly suggests that the claudin-low classifier may be measuring the amount of tumor associated normal stroma tissue within each of the subtypes. Other than a classifier built on cell line models there is no direct evidence that this subtype is not driven by normal cells and the data presented in this manuscript could account for why this subtype occurs across other subtypes and varies in composition with differing datasets. Further analysis could shed light on these findings and

A. These data presented in the manuscript are compelling and analysis of the H&E sections from claudin low tumors may provide further insight from METABRIC and TCGA (these are available).

B. Also subtyping of paired tumor and adjacent normal tissue could reveal similar conclusions. There are several available in the TCGA and there are likely additional publicly available datasets on GEO or SRA. A quick analysis of TCGA that includes 1095 tumor and 113 normal adjacent provides interesting clues as the number of claudin-low tumors increases from 4% to 40% when only tumors are normalized together versus when tumor and adjacent normal samples are

normalized together. Furthermore, greater than 80% of the matched normal are predicted as claudin-low and majority of the paired tumor is classified as other subtype. The authors should consider doing a more throughout analysis in the TCGA that include normal or normal adjacent breast and evaluating tumor cellularity in the comparisons.

C. It would be nice to see the claudin-low score plotted vs. tumor cellularity in a scatterplot for each of the tumors that have the data available. If not plot claudin-low score vs. Estimate stromal score.

D. Clearly there are cell-lines that are claudin-low and have undergone EMT. Some of those cell lines were derived from cancers with unique histological distinctions such as metaplastic. However, the gene expression changes that accompany EMT are likely very similar to normal stroma and the claudin-low classifier may identify both tumors with EMT features and tumors with more normal tumor associated stroma. Perhaps a classifier could be trained using tumors with metaplastic features in the H&E to tease out these tumors from those that have high amounts of normal stroma rather than choosing 19 pathognomonic genes. A true claudin-low signature should be able to identify tumors with EMT features independent of the amount of normal stroma and not classify nearly all the adjacent normal tissue as claudin-low.

Minor Concerns

1. Figure 3 would be improved with annotations to histological subtype (IDC, ILC, metaplastic, medullary). These are available in the METABRIC dataset. It would be nice to know whether the claudin-low tumors with metaplastic features are enriched in a certain intrinsic subtype/
2. Please include a supplemental file metadata of samples analyzed that include sample identifier, subtype, claudin-low classification and others.
3. The authors created a condensed claudin-low gene list (Supplementary Table S3), consisting of 19 genes representing only the pathognomonic gene expression characteristics of claudin-low tumors: Low expression of cell-cell adhesion genes, high expression of epithelial-mesenchymal transition genes, and gene expression patterns typical of stem cell-like/less differentiated cells. More justification is needed for choosing these genes or an different unbiased approach should be used that can differentiate normal stoma.
4. The main difference between the core- and other- claudin low predictors is that the core-predictor filters out many basal-like tumors. This is likely because the other-claudin low signature was generated from 9 cell lines (all basal) and does not include ER positive tumors with EMT features (not sure if there is much evidence for these tumors as most metaplastic/anaplastic tumor are TNBC). The other possibility is that the core-claudin low signature is identifying ER+ tumors with low cellularity and high tumor associated stroma. The authors should evaluate H&E sections from each of the claudin low tumors identified by each signature in METABRIC and TCGA.

Reviewer #2 (Remarks to the Author):

The article by Fougner and colleagues evaluated the genomic, transcriptomic, and clinical features of Claudin-low breast cancers. Specifically, they evaluated whether Claudin-low tumors comprise a single homogenous entity or a complex heterogenous group of tumors that permeate all other subtypes. They have also performed a re-definition of these tumors by the use of a reduced gene-set that was manually selected to reflect some of the published characteristics of claudin-low tumors. They have done this by identifying claudin-low tumors (N = 87) in the METABRIC cohort, with attempts at validation in the Oslo2 and TCGA cohorts. Overall, this study provides a comprehensive evaluation of claudin-low tumors, particularly within the context of three of the major intrinsic breast cancer subtypes (Basal-like, Luminal A, and Normal-like). The authors have

done a good job in presenting their results within the context of the literature in the field and have demonstrated some exciting findings. Of note, they have shown that claudin-low tumors are heterogeneous and often demonstrate biological behavior that mimics their parent intrinsic subtype. This finding is important and calls into question the current practice of overwriting the PAM50 subtypes if a determination of claudin-low phenotype is made on the same tumor sample. Nevertheless, unanswered questions abound in this manuscript that need to be addressed.

Major comments:

i. A major issue that needs to be addressed head-on is the observed strong inverse correlation between the prevalence of claudin-low tumors and tumor cellularity (Figure 5). The authors noted a prevalence of claudin-low tumors of 3.0% in the TCGA cohort, 4.6% in the METABRIC cohort, and 7.6% in the Oslo2 cohort, with corresponding tumor cellularity cutoff-points of 60%, 40%, and none in these cohorts, respectively. This raises an important concern regarding whether claudin-low tumors are indeed a tumor subtype or an artefact of normal cell infiltration within tumors. This question is also relevant in light of the key finding of this study, wherein the claudin-low phenotype was observed within the different breast cancer subtypes. It is unclear how much of an individual tumor subtype's "claudin-lowness" can be attributed to its intrinsic biology or to the extent of "normal" cell contamination of the tumor sample. Additionally, since it is unlikely for tumor sampling to capture 100% of the tumor cells (and completely exclude normal cell infiltrates), there is the need to rule out intratumor heterogeneity as a cause of this finding – or to at least acknowledge the inability to do so as a major limitation of this manuscript and any inferences that can be drawn from its findings.

ii. With sample sizes of 45, 28, and 9 for Basal-like/claudin-low, Normal-like/claudin-low and Luminal A-like/claudin-low, it is doubtful that the authors had sufficient power to detect differences in survival between tumors stratified by claudin-low status in these subtypes. It would have been helpful if the authors provided the number of breast cancer-specific deaths for the comparison groups. A "number at risk" table within the KM survival curves could also suffice.

iii. There was no discussion on the limitations of this study. Inadequate sample size, particularly for survival analysis, inability to rule out intratumor heterogeneity, potential lack of external generalizability, and lack of morphology data are a few possible limitations that warrant further discussions in the manuscript.

The below are also offered for consideration:

iv. The authors make a case for a re-definition of claudin-low tumors based on their observation that, as currently defined, the behavior of claudin-low tumors closely resembles that of their parent intrinsic subtype. They have also found some differences between claudin-low and non-claudin-low tumors of the same intrinsic subtype. Fundamentally, however, they did not find differences in clinical outcomes between claudin-low and non-claudin-low tumors of the same intrinsic subtype. What is not clear throughout the manuscript is how the proposed re-definition of claudin-low tumors addresses these issues thereby making the significance/importance of their proposed new classifier questionable.

To be more specific:

- Apart from basal-like tumors, the proposed new classifier did not result in any substantial re-classification of claudin-low tumors within the other intrinsic subtypes. Further, comparison of Figure 1 and Fig.S6 show that the genomic and transcriptomic characteristics of the CoreCL phenotype tracks closely with that of its parent intrinsic subtype (particularly for Luminal A and Normal-like tumors) similar to what they observed for claudin-low tumors in general. It is therefore unclear to me how much the new classification departs from the nine-cell line predictor overall.

- It would have been interesting to see how the survival curves for the Normal-like/CoreCL, Luminal A/CoreCL and Basal-like/CoreCL differed from one another like it was shown in Figure 2 for claudin/low tumors. I hope that the authors can show this as it would help clarify the extent of homogeneity between these phenotypes.

v. The major difference between the current nine-cell line-based classifier and the proposed re-definition of the claudin-low phenotype was seen in the Basal-like subtype. However, the Basal-like subtype is a heterogeneous entity comprised of different histological subtypes. For instance, medullary carcinomas are predominantly Basal-like (triple-negative) tumors that are characterized by their distinctive high immune infiltration. Are the authors able to ascertain how much of the differences between Basal-like CoreCL and Basal-like OtherCL is driven by histology? This is especially relevant given that their proposed classifier was designed to exclude tumors with marked immune and stromal cell infiltration – features that were considered non-pathognomonic of claudin-low tumors by the authors. It'll be reassuring to know that the proposed classifier is not overly excluding immune-rich histological subtypes of breast cancer.

vi. High immune gene expression has been consistently described in the literature as a feature of claudin-low tumors. In fact, the authors cited this immune repertoire as being relevant for the efficacy of chemo and immunotherapies in claudin-low tumors in their discussion. I am curious to understand how they align this statement with the fact that their condensed claudin-low gene list excluded immune gene expression by focusing only on gene expression characteristics related to cell-cell adhesion, epithelial-mesenchymal transition, and stem cell-like/less differentiation.

vii. In the METABRIC cohort, the majority of Other-CL tumors were Basal-like with very little contribution from Luminal A and Normal-like. This was however different in the Oslo2 cohort where a substantial proportion of the Other-CL tumors were Luminal A. What could be responsible for these differences?

Reviewer #3 (Remarks to the Author):

This is a well written paper by Fougner and colleagues, in which they propose that Claudin-low breast cancer is not a unique subtype of breast cancer but rather an element or other dimension that can exist in breast cancers of all intrinsic subtypes albeit more prevalent in basal-like tumours.

Using an in silico approach they identify 87 claudin-low tumours in the Metabric data base using the 9 cell line classifier of Prat and colleagues. The majority (51.7%) of the claudin-low tumours so identified were basal, 32.2% normal-like and 10.3% luminal A. Then, through a series of comparisons they demonstrate that the claudin-low phenotype of each sub-group is reflective of their intrinsic subtype; including ER expression, lower mutational burden, lower genomic instability, driver mutations, proliferation etc.. In addition, they demonstrate that the survival of claudin-low tumours is not inferior to that of non-claudin low tumours of the same intrinsic subtype.

The authors then refine, again by in silico techniques the gene signature of claudin low tumours to a core 19 gene list and apply this new classifier to the Metabric database. This approach yielded 79 'core' claudin low tumours that are predominantly basal-like and yet have sustained and recognisable differences from the non-claudin low basal-like tumours including lower proliferation, lower mutational burden, increased genomic stability and increased stromal cell and immune cell infiltrate. This new classifier was subsequently validated in 2 different breast cancer databases. The value of this work is that it does validate the existence of the claudin-low phenotype with some evidence to suggest its predominantly a group of basal-like breast cancers with immune infiltrate. The latter may have clinical implications.

We would like to thank the reviewers for enthusiastic and insightful comments regarding our manuscript. We have addressed all comments and revised the manuscript as described in detail below.

Reviewer #1

In this manuscript the authors investigate genomic, transcriptomic, and clinical features of claudin-low breast tumors. The claudin-low breast cancer subtype that was initially identified by clustering human tumors and murine breast cancer (BRCA) models and then a classifier built using human cell lines to minimize immune and stromal infiltration. It is characterized by low expression of cell-cell adhesion genes, high expression of epithelial-mesenchymal transition (EMT) genes, and stem cell-like/less differentiated gene expression patterns.

The authors have identified 87 claudin-low tumors from the METABRIC cohort and performed analysis of ER, mutation burden, genomic instability, MKI67 and specific oncogenic mutations within the intrinsic subtypes. Within each of the intrinsic subtypes, claudin-low tumors were distinguished by low genomic instability, mutational burden and proliferation levels, and high levels of immune and stromal cell infiltration. The authors show no evidence that claudin-low cancers are an indicator of poor prognosis when binned by intrinsic subtype, as previously described by others when pooled together. They provide compelling evidence that claudin-low is not simply a subtype as previously portrayed, but is a complex additional phenotype which span various intrinsic subtypes. They have developed an alternative method for identifying claudin-low tumors using a biased gene set and uncovered potential weaknesses in the established claudin-low classifier.

These findings suggest heterogeneity in claudin-low breast tumors, and substantiate a re-definition of claudin-low as a cancer phenotype. They also provide a reasonable argument that tumor cellularity contributes to the varying degrees of claudin-low-tumors (1.5-14%) between differing datasets. Given the lower mutation, decreased genomic instability, high percentage of stroma (by ESTIMATE) and the varying amounts by tumor cellularity, these data also suggest that perhaps a portion of claudin-low tumors are likely tumors from any intrinsic subtype with a high proportion of normal stroma. The authors should consider additional studies of H&E sections along with histopathological characteristics of tumors such as metaplastic features in defining the subtype. The potential confounding issue of normal stroma could be demonstrated with additional evaluation of patient matched tumor and adjacent normal tissue.

Major Concerns

1. The fact that a large proportion of normal-like subtype are claudin-low, combined with the fact the claudin low occur across all subtypes, with lower genomic instability, copy number and mutational burden, and the number of claudin-low tumors is dependent on tumor cellularity

strongly suggests that the claudin-low classifier may be measuring the amount of tumor associated normal stroma tissue within each of the subtypes. Other than a classifier built on cell line models there is no direct evidence that this subtype is not driven by normal cells and the data presented in this manuscript could account for why this subtype occurs across other subtypes and varies in composition with differing datasets. Further analysis could shed light on these findings and

A. These data presented in the manuscript are compelling and analysis of the H&E sections from claudin low tumors may provide further insight from METABRIC and TCGA (these are available).

METABRIC

Tumors in the METABRIC cohort were gathered from 1977 through 2005, and from five different centers in the UK and Canada. The original annotations released with the METABRIC cohort were primary pathology reports, and these descriptions are therefore not consistent over time and between centers. A central pathology review of the METABRIC cohort was therefore carried out and published in 2018¹. These data were kindly made available to us by Dr. Elena Provenzano (Addenbrookes Hospital, Cambridge). We use the original METABRIC annotation data for tumor cellularity/purity as this was not evaluated in the central pathology review.

Of the 1886 tumors used in our study, 1575 could be evaluated in the central pathology review. Histological evaluations were available for 71 of 87 claudin-low tumors (nine-cell line) and 66 of 79 CoreCL tumors. The distribution of histological classifications was as follows:

Histology	Claudin-low (n = 71)	CoreCL (n = 66)	Entire cohort (n = 1575)
NST	50 (70%)	42 (64%)	1168 (74%)
Lobular	6 (8%)	9 (14%)	108 (7%)
Mixed NST/Special type	6 (8%)	8 (12%)	185 (12%)
Medullary-like	3 (4%)	1 (2%)	22 (1%)
Tubular	3 (4%)	4 (6%)	25 (2%)
Mucinous	1 (1%)	1 (2%)	22 (1%)
Special types	2 (3%)	1 (2%)	45 (3%)

Sum of percentages may not equal 100 due to rounding

NST = No special type

The histological classifications of claudin-low tumors in the METABRIC cohort did not differ notably from the histological classifications of the cohort as a whole. In the entire cohort, only one tumor was classified as metaplastic (categorized under special types). This metaplastic tumor was classified as claudin-low by the nine-cell line predictor, and also as CoreCL.

Where appropriate, pathologists also provided unstructured comments alongside histological classifications; no notable/recurring comments were made regarding claudin-low tumors. No trends were observed when the histological classifications of claudin-low tumors were stratified by intrinsic subtype.

For those tumors with tumor cellularity data available, the distribution was as follows:

Tumor cellularity	Claudin-low (n = 77)	CoreCL (n = 71)	Entire cohort (n = 1833)
High	19 (25%)	17 (24%)	932 (51%)
Moderate	39 (51%)	31 (44%)	706 (39%)
Low	19 (25%)	23 (32%)	195 (11%)

Sum of percentages may not equal 100 due to rounding

On average, claudin-low tumors were more likely to have low or moderate tumor cellularity than the rest of the cohort. However, only 25% of all claudin-low tumors showed low tumor cellularity, and of all tumors with low tumor cellularity, only 10% were claudin-low. Similarly, 32% of CoreCL tumors had low tumor cellularity, and CoreCL tumors only accounted for 12% of all low-cellularity tumors.

No trends were observed when the cellularity of claudin-low tumors was stratified by intrinsic subtype.

Oslo2

Due to issues surrounding the European General Data Protection Regulation (GDPR), we are unable to publish the histopathological classifications of individual tumors from Aure et al.² (including tumor cellularity). There is however only one tumor with metaplastic morphology in the Oslo2 cohort². Metaplastic tumors can therefore at most represent 3-4% of claudin-low tumors in the dataset.

TCGA

The distribution of histological classifications in the TCGA cohort were as follows:

Histology	Claudin-low (n = 32)	Entire Cohort (n = 1082)
NST	16 (50%)	776 (72%)
Lobular	7 (22%)	201 (19%)
Mixed/NST	1 (3%)	29 (3%)
Medullary-like	0	6 (1%)
Mucinous	0	17 (2%)
Metaplastic	6 (19%)	8 (1%)
Other	2 (6%)	45 (4%)

Sum of percentages may not equal 100 due to rounding

With the exception of metaplastic tumors, claudin-low tumors generally showed similar histology to that observed in the cohort as a whole. There were 8 metaplastic tumors in the cohort, of which 6 were classified as claudin-low. Thus, metaplastic tumors were significantly overrepresented in the set of claudin-low tumors ($P < 0.001$, Fisher's exact test), but metaplastic tumors also only accounted for a 19% of all claudin-low tumors.

No trends were observed when the histology of claudin-low tumors was analyzed in a subtype-specific manner.

By histological evaluation, claudin-low tumors had a mean of 67% tumor cells, compared to a mean of 74% for non-claudin-low tumors. Claudin-low tumors had a mean stromal infiltration of 26%, compared to a mean of 21% for non-claudin-low tumors. These corroborate our findings from ESTIMATE indicating that claudin-low tumors have higher levels of stromal infiltration than non-claudin-low tumors. There were however 316 tumors in the cohort with stromal infiltration greater than the mean level of stromal infiltration in claudin-low tumors. Only 14 of these (4%) were classified as claudin-low. Similarly, for cellularity, only 5% of tumors with a tumor cell percentage less than the average tumor cell percentage in claudin-low tumors were in fact classified as claudin-low.

In sum, the histological data from these three cohorts confirm that claudin-low tumors have high levels of non-tumor cell infiltration (in line with findings from ESTIMATE), and that most metaplastic tumors are classified as claudin-low. However, of all tumors with high levels of stromal and immune cell infiltration, only a minority are classified

as claudin-low, and metaplastic tumors only account for a small proportion of all claudin-low tumors. Thus, neither stromal/immune cell infiltration, nor histological subtype, can adequately explain the claudin-low gene expression phenotype. We have added a brief summary of histological classifications to the updated manuscript in lines 87-89, and 280 - 283.

B. Also subtyping of paired tumor and adjacent normal tissue could reveal similar conclusions. There are several available in the TCGA and there are likely additional publicly available datasets on GEO or SRA. A quick analysis of TCGA that includes 1095 tumor and 113 normal adjacent provides interesting clues as the number of claudin-low tumors increases from 4% to 40% when only tumors are normalized together versus when tumor and adjacent normal samples are normalized together. Furthermore, greater than 80% of the matched normal are predicted as claudin-low and majority of the paired tumor is classified as other subtype. The authors should consider doing a more throughout analysis in the TCGA that include normal or normal adjacent breast and evaluating tumor cellularity in the comparisons.

The reviewer's observations are intriguing, and from what we gather, the concern the reviewer is raising is that claudin-lowness (as measured by the nine-cell line predictor) might be heavily influenced by stromal infiltration. This is a vital consideration, and this point is directly demonstrated by the reviewer's suggestion in point 1C. We also approach this question in 1A & 1D, as well as discussing the question at large in response to Reviewer #2, point i. We hope that these responses adequately address this concern.

C. It would be nice to see the claudin-low score plotted vs. tumor cellularity in a scatterplot for each of the tumors that have the data available. If not plot claudin-low score vs. Estimate stromal score.

We thank the reviewer for the astute suggestion, which revealed the following relationship in the METABRIC cohort:

Left: ESTIMATE stromal score vs. distance to the claudin-low centroid from the nine-cell line predictor. Data points colored by intrinsic subtype and claudin-low status, as in the manuscript: claudin-low = yellow, basal-like = red, HER2-enriched = pink, LumA = dark blue, LumB = light blue, normal-like = green. Claudin-low tumors identified using the nine-cell line predictor.

Right: CoreCL tumors colored yellow. Data otherwise identical to left panel.

There is a startlingly strong correlation between stromal score (from ESTIMATE) and distance to the claudin-low centroid ($R^2 = 0.76$). This lends support to our observation that the nine-cell line predictor may often identify tumors as claudin-low primarily due to their stromal infiltration.

The reviewers have, directly and indirectly, raised the question: could claudin-low simply be an artefact of stromal infiltration? In the context of this question, it is concerning to observe that the nine-cell line predictor primarily measures stromal infiltration, and that tumors classified as claudin-low are those with the highest levels of stromal infiltration. To address this question directly though, we can attempt to identify claudin-low tumors solely based on StromalScore, for example by setting a cut-off at a level such that 80% of claudin-low tumors have a StromalScore above the cut-off. Using the nine-cell line classification of claudin-low, this cut-off would be placed at StromalScore = 1117, and for the CoreCL classification, the cut-off would be set at StromalScore = 1246. By definition, this cut-off would have a sensitivity of 80%.

However, of all tumors with StromalScore greater 1117, only 22% are classified as claudin-low by the nine-cell line predictor. Of all tumors with StromalScore greater than 1246, only 34% are classified as CoreCL.

As an additional observation, we draw attention to four claudin-low outlier tumors: one near $x = -750$, $y = 65$, and a group of three tumors around $x = 500$, $y = 50$. It is interesting to note that these four tumors have relatively low StromalScore and a high distance from the claudin-low centroid, yet they are classified as claudin-low by both the nine-cell line predictor and the CoreCL method. Their classification as claudin-low does therefore not appear to be a result of stromal infiltration, and as these tumors are CoreCL, their claudin-low status cannot simply be ascribed to extremely poor correlation to the nine-cell line other centroid.

In sum, the above figures indicate that the nine-cell line claudin-low centroid may essentially be a measure of stromal infiltration. This observation strengthens the arguments we make regarding the limitations of the nine-cell line predictor. However, a hard cut-off for stromal infiltration does not accurately identify claudin-low tumors, by either classification of claudin-low used in this study. Therefore, while stromal infiltration is a very strong gene expression signal in claudin-low tumors (which partially overlaps with the EMT-like gene expression signature³), it does not by itself explain a tumor's claudin-lowness. In other words, the vast majority of claudin-low tumors have high degrees of stromal infiltration, but only a subset of tumors with marked stromal infiltration are claudin-low. These gene expression-based findings are in line with the histological results discussed in point 1A.

We have included the above left chart in Supplementary Fig. 3, along with analogous figures for the following relationships:

- *Distance to Other centroid ~ Stromal score*
- *Distance to claudin-low centroid ~ Immune score*
- *Distance to Other centroid ~ Immune score*

These figures are referred to in lines 183-187 of the updated manuscript, and their implications are discussed in lines 232 – 234, 357 – 369 and 407 – 416.

In light of the above findings, we would also like to take the opportunity to briefly compare the genomic stability of nine-cell line claudin-low tumors and CoreCL tumors. We show genomic instability index and mutational burden in these groups in Fig. 1 and Supplementary Fig. 7. Given the above findings, and the reviewers' concerns, it is however also pertinent to present the same data with stromal infiltration taken into account.

In the following figures, we only show data for tumors with StromalScore > 1117 (nine-cell line claudin-low; left) and StromalScore > 1246 (CoreCL; right):

These visualizations show that claudin-low tumors have lower GII and mutational burden than non-claudin-low tumors, also when stromal infiltration is taken into consideration. These findings support our claim that CoreCL tumors are more genomically stable than the claudin-low tumors identified by the nine-cell predictor, also when stromal infiltration is taken into account.

D. Clearly there are cell-lines that are claudin-low and have undergone EMT. Some of those cell lines were derived from cancers with unique histological distinctions such as metaplastic. However, the gene expression changes that accompany EMT are likely very similar to normal stroma and the claudin-low classifier may identify both tumors with EMT features and tumors with more normal tumor associated stroma. Perhaps a classifier could be trained using tumors with metaplastic features in the H&E to tease out these tumors from those that have high amounts of normal stroma rather than choosing 19 pathognomonic genes. A true claudin-low signature should be able to identify tumors with EMT features independent of the amount of normal stroma and not classify nearly all the adjacent normal tissue as claudin-low.

We acknowledge the issue of stromal infiltration in claudin-low tumors, and the reviewer's suggestion of making a classifier based on metaplastic tumors is interesting. However, we believe this approach has certain important drawbacks.

We show in point 1A that metaplastic tumors only account for a small proportion of all claudin-low tumors (by either classifier used in our study). In the METABRIC cohort metaplastic tumors account for no more than 1-2% of claudin-low tumors. Therefore, training a claudin-low classifier on metaplastic tumors would heavily bias the classifier towards the characteristics of a minor subset of claudin-low tumors. For example, most metaplastic tumors are triple negative and/or basal-like^{4,5}. We have however shown that a major proportion of claudin-low tumors express ER (METABRIC: 58% of nine-cell line claudin-low tumors, 72% of CoreCL tumors). An unbiased metaplastic vs. all others classifier would therefore be likely to overly exclude luminal and normal-like claudin-low tumors. If one argues from the position that metaplastic tumors are the only genuine claudin-low tumors, this bias might be reasonable. We, however, believe that we have presented robust evidence that there exists a set of breast tumors with EMT-like genomic⁶ and transcriptomic traits, which cannot be attributed to metaplastic histology or stromal infiltration.

When optimizing a claudin-low classifier, what exactly are we optimizing for? The nine-cell line predictor was optimized for replicating findings from hierarchical clustering⁷. As we show in response to point 1C (and mention in the manuscript), this unbiased approach yielded a classifier which essentially measures stromal infiltration. This highlights how powerful the gene expression signal from stromal infiltration is in claudin-low tumors. However, since claudin-low tumors were originally characterized, EMT has emerged as the unifying feature for the tumor group^{6,8,9}. The EMT-like gene expression signature in claudin-low tumors seems to be more subtle than the partially overlapping³ gene expression signature of stromal tissue, which is why we chose a biased approach to building a new classification method. We therefore optimized for EMT-like features, and in line with the findings of Morel et al.⁶, this identified a more genomically stable set of tumors (as illustrated in Supplementary Fig. 7). However, with regards to building a classifier based on metaplastic tumors, the classifier would be optimizing for "metaplastic-ness". As we discuss above, metaplastic tumors are not representative of claudin-low tumors as a whole, and it is therefore unclear to us how relevant this would be for our study. Additionally, metaplastic tumors show high levels of stromal infiltration, similar to that seen in non-metaplastic claudin-low tumors. It is therefore likely that an unbiased metaplastic classifier would have the same issue as presented for the nine-cell line predictor in our response to point 1C.

Ultimately, we can only evaluate how good a classifier is in the context of the question: Good for what? Claudin-low status does not appear to influence survival, and we can therefore not evaluate a classifier by its prognostic value. There are no currently available targeted therapies against claudin-low features, and we can therefore not evaluate a classifier for its ability to predict treatment response. There are no biological

gold standards defining claudin-low, such as histology or protein markers, which a classifier could be benchmarked against. Therefore, we cannot claim that our method is “better” than the nine-cell line predictor. Rather, we present evidence that our method is more appropriate for identifying tumors defined by EMT-like gene expression patterns and genomic stability. We emphasize that the identification of claudin-low tumors is not a solved problem, and that our method should principally be viewed as an additional tool in the toolkit for studying claudin-low tumors.

We include a subset of these deliberations, and a discussion of the limitations to our approach, in lines 357 – 369 and 407 – 416. Additional relevant discussion can also be found in lines 325 - 342.

Minor Concerns

1. Figure 3 would be improved with annotations to histological subtype (IDC, ILC, metaplastic, medullary). These are available in the METABRIC dataset. It would be nice to know whether the claudin-low tumors with metaplastic features are enriched in a certain intrinsic subtype.

We have added the requested annotations to Figure 3:

As discussed in point 1A, only one tumor in the METABRIC cohort showed metaplastic morphology¹. We do not observe any subtype-specific histological trends in this analysis.

2. Please include a supplemental file metadata of samples analyzed that include sample identifier, subtype, claudin-low classification and others.

The requested data has been included as supplemental files for each cohort:

- *METABRIC: Supplementary Table 1*
- *Oslo2: Supplementary Table 4*
- *TCGA: Supplementary Table 5*

3. The authors created a condensed claudin-low gene list (Supplementary Table S3), consisting of 19 genes representing only the pathognomonic gene expression characteristics of claudin-low tumors: Low expression of cell-cell adhesion genes, high expression of epithelial-mesenchymal transition genes, and gene expression patterns typical of stem cell-like/less differentiated cells. More justification is needed for choosing these genes or an different unbiased approach should be used that can differentiate normal stoma.

Please see our response to point 1D, addressing our reasoning for using a biased predictor. In brief, we deliberately chose a biased approach due to observations related to the issues discussed in our responses to point 1C and Reviewer #2 point i. We have added more justification for our decisions when building this classifier (see lines 407 – 416), and emphasized the limitations and biases associated with our approach (see lines 357 – 369).

4. The main difference between the core- and other- claudin low predictors is that the core-predictor filters out many basal-like tumors. This is likely because the other-claudin low signature was generated from 9 cell lines (all basal) and does not include ER positive tumors with EMT features (not sure if there is much evidence for these tumors as most metaplastic/anaplastic tumor are TNBC). The other possibility is that the core-claudin low signature is identifying ER+ tumors with low cellularity and high tumor associated stroma. The authors should evaluate H&E sections from each of the claudin low tumors identified by each signature in METABRIC and TCGA.

We perform a systematic evaluation of H&E findings in response to 1A, in which we find no subtype subtype-specific differences in histology or tumor cellularity in METABRIC or TCGA. Additionally, we were able to access 21 H&E tissue sections from claudin-low tumors in the Oslo2 cohort, and we enlisted the help of a pathologist who kindly evaluated these sections. This evaluation confirmed that claudin-low tumors showed high degrees of stromal infiltration, and that only a small subset show

metaplastic histology. Stromal tissue infiltrated the tumors and was not a sampling artefact. No subtype specific variations were observed.

Reviewer #2

The article by Fougner and colleagues evaluated the genomic, transcriptomic, and clinical features of Claudin-low breast cancers. Specifically, they evaluated whether Claudin-low tumors comprise a single homogenous entity or a complex heterogenous group of tumors that permeate all other subtypes. They have also performed a re-definition of these tumors by the use of a reduced gene-set that was manually selected to reflect some of the published characteristics of claudin-low tumors. They have done this by identifying claudin-low tumors (N = 87) in the METABRIC cohort, with attempts at validation in the Oslo2 and TCGA cohorts. Overall, this study provides a comprehensive evaluation of claudin-low tumors, particularly within the context of three of the major intrinsic breast cancer subtypes (Basal-like, Luminal A, and Normal-like). The authors have done a good job in presenting their results within the context of the literature in the field and have demonstrated some exciting findings. Of note, they have shown that claudin-low tumors are heterogeneous and often demonstrate biological behavior that mimics their parent intrinsic subtype. This finding is important and calls into question the current practice of overwriting the PAM50 subtypes if a determination of claudin-low phenotype is made on the same tumor sample.

Nevertheless, unanswered questions abound in this manuscript that need to be addressed.

Major comments

i. A major issue that needs to be addressed head-on is the observed strong inverse correlation between the prevalence of claudin-low tumors and tumor cellularity (Figure 5). The authors noted a prevalence of claudin-low tumors of 3.0% in the TCGA cohort, 4.6% in the METABRIC cohort, and 7.6% in the Oslo2 cohort, with corresponding tumor cellularity cutoff-points of 60%, 40%, and none in these cohorts, respectively. This raises an important concern regarding whether claudin-low tumors are indeed a tumor subtype or an artefact of normal cell infiltration within tumors. This question is also relevant in light of the key finding of this study, wherein the claudin-low phenotype was observed within the different breast cancer subtypes. It is unclear how much of an individual tumor subtype's "claudin-lowness" can be attributed to its intrinsic biology or to the extent of "normal" cell contamination of the tumor sample. Additionally, since it is unlikely for tumor sampling to capture 100% of the tumor cells (and completely exclude normal cell infiltrates), there is the need to rule out intratumor heterogeneity as a cause of this finding – or to at least acknowledge the inability to do so as a major limitation of this manuscript and any inferences that can be drawn from its findings.

We thank the reviewer for highlighting this important issue and agree that it should be discussed more prominently in our manuscript. We would like to raise the following

arguments to address the question of whether claudin-low might be an artefact of normal-cell infiltration:

First, we have found that the distribution of intrinsic subtypes within the set of claudin-low tumors differs significantly from the distribution of intrinsic subtypes for non-claudin-low tumors ($P < 0.001$ in the METABRIC cohort, χ^2 -test). The claudin-low phenotype is significantly overrepresented in basal-like and normal-like tumors, and significantly under-represented in HER2-enriched, LumA and LumB tumors ($P = 0.001$ for HER2-enriched, $P < 0.001$ for all other, Fisher's exact test). If claudin-low were purely a sampling artefact from tumor biopsies, irrelevant to a tumor's biology, one would expect this sampling artefact to affect all intrinsic subtypes similarly. As this is not the case, we would argue that stromal infiltration in claudin-low tumors is biologically real and not simply a sampling artefact. It may be that basal-like and normal-like tumors carry some characteristic which makes a stromally infiltrated tumor sampling more likely, but this would be indicative of a genuine biological feature.

Second, if claudin-low is simply an artefact of normal-cell infiltration, should normal-cell infiltration not be highly predictive of claudin-low status? We discuss this question in response to Reviewer #1, points 1A & 1C, and ask that these responses are taken into consideration at this point. In brief, we show that claudin-low tumors almost always carry high levels of stromal and immune cell infiltration. However, of all tumors with levels of normal-cell infiltration similar to that seen in claudin-low tumors, only a minority are in fact classified as claudin-low. Normal-cell infiltration could therefore be described as "necessary" for claudin-lowness, but not sufficient.

Finally, we would like to highlight the wealth of previous studies of claudin-low tumors. For example, studies have shown, through immunohistochemical staining, that claudin-low tumor cells express protein-profiles in line with the gene expression characteristics identified in bulk claudin-low tumors^{7,9,10}. There exist numerous cell-lines with claudin-low features, which cannot be attributed to special histological subtypes^{7,11}. The work of Morel et al.^{6,9} provides a mechanistic rationale for the lower mutational burden and genomic instability in claudin-low tumors. These previously published findings represent a small subset of the evidence showing that claudin-low is an EMT-like phenotype, which, to some extent, must be attributable to the inherent characteristics of claudin-low tumor cells

In sum, these arguments indicate that claudin-low cannot simply be considered an artefact of normal-cell infiltration. Non-tumor-cell infiltration is undoubtedly a significant feature in these tumors, and we agree with the reviewer that it is difficult to ascertain the extent to which this drives the characteristics of bulk tumor samples. This is an important limitation of our study, and we mention in our initial submission that single cell sequencing will likely be required to disentangle the characteristics of tumor and non-tumor cells. This point is emphasized further in the updated manuscript. It must, however, also be mentioned that countless studies have demonstrated the impact of microenvironmental influences in cancer and it is highly unlikely that the infiltrating

normal-cells in claudin-low tumors are biologically inert¹²⁻¹⁴. The significance of non-tumor cells in claudin-low tumors should therefore not be discounted.

We have added a discussion surrounding this issue in lines 325 - 342 and 357 - 361.

ii. With sample sizes of 45, 28, and 9 for Basal-like/claudin-low, Normal-like/claudin-low and Luminal A-like/claudin-low, it is doubtful that the authors had sufficient power to detect differences in survival between tumors stratified by claudin-low status in these subtypes. It would have been helpful if the authors provided the number of breast cancer-specific deaths for the comparison groups. A “number at risk” table within the KM survival curves could also suffice.

We agree that this is an important consideration when discussing the absence of significant differences in survival in claudin-low versus non-claudin-low tumors. The number of breast cancer specific deaths have been added to the figure legends for the Kaplan-Meier curves (Fig. 2, Fig. 5, Supplementary Fig. 8). Discussion regarding sample size has been added in lines 352 – 357.

iii. There was no discussion on the limitations of this study. Inadequate sample size, particularly for survival analysis, inability to rule out intratumor heterogeneity, potential lack of external generalizability, and lack of morphology data are a few possible limitations that warrant further discussions in the manuscript.

We thank the reviewer for pointing out this oversight in our paper. We have added a discussion on the limitations of the study in lines 352 - 369 and hope that this adequately addresses the concern.

The below are also offered for consideration:

iv. The authors make a case for a re-definition of claudin-low tumors based on their observation that, as currently defined, the behavior of claudin-low tumors closely resembles that of their parent intrinsic subtype. They have also found some differences between claudin-low and non-claudin-low tumors of the same intrinsic subtype. Fundamentally, however, they did not find differences in clinical outcomes between claudin-low and non-claudin-low tumors of the same intrinsic subtype. What is not clear throughout the manuscript is how the proposed re-definition of claudin-low tumors addresses these issues thereby making the significance/importance of their proposed new classifier questionable.

To be more specific:

- Apart from basal-like tumors, the proposed new classifier did not result in any substantial re-classification of claudin-low tumors within the other intrinsic subtypes.

Further, comparison of Figure 1 and Fig.S6 show that the genomic and transcriptomic characteristics of the CoreCL phenotype tracks closely with that of its parent intrinsic subtype (particularly for Luminal A and Normal-like tumors) similar to what they observed for claudin-low tumors in general. It is therefore unclear to me how much the new classification departs from the nine-cell line predictor overall.

- It would have been interesting to see how the survival curves for the Normal-like/CoreCL, Luminal A/CoreCL and Basal-like/CoreCL differed from one another like it was shown in Figure 2 for claudin/low tumors. I hope that the authors can show this as it would help clarify the extent of homogeneity between these phenotypes.

To clarify, the “re-definition” of claudin-low which we aim to achieve pertains to the question of whether claudin-low is a phenotype or an intrinsic subtype (as it has previously been depicted). This re-definition, as summarized in Fig. 6, applies to claudin-low tumors irrespective of whether they are identified using our method (CoreCL) or the nine-cell line predictor. Therefore, we do not make any claim that CoreCL tumors are independent of their intrinsic subtype, nor that their prognosis differs from non-claudin-low tumors. Rather, we claim that claudin-low tumors identified using the condensed gene list are more consistently characterized by EMT-like features, while tumors identified using the nine-cell line predictor are more strongly defined by stromal infiltration (see response to Reviewer #1, points 1C & 1D). Contemporary studies indicate that EMT-like features are key to claudin-lowness^{6,8,9,15}, and we therefore consider our method advantageous. We hope that this answers the first part of the concern.

The survival curves for CoreCL tumors are shown:

In line with the survival curves for claudin-low tumors identified by the nine-cell line predictor, we find no evidence of CoreCL status affecting survival. The difference in survival between basal-like CoreCL, LumA CoreCL and normal-like CoreCL tumors does not approach statistical significance ($P = 0.32$). We ascribe this to the reduction in sample

size for the basal-like tumor group (basal-like CoreCL $n = 25$; basal-like nine-cell line claudin-low $n = 45$), which accounts for the majority of disease specific deaths.

We have added the above figure as a supplementary file (Supplementary Fig. 8)

v. The major difference between the current nine-cell line-based classifier and the proposed re-definition of the claudin-low phenotype was seen in the Basal-like subtype. However, the Basal-like subtype is a heterogeneous entity comprised of different histological subtypes. For instance, medullary carcinomas are predominantly Basal-like (triple-negative) tumors that are characterized by their distinctive high immune infiltration. Are the authors able to ascertain how much of the differences between Basal-like CoreCL and Basal-like OtherCL is driven by histology? This is especially relevant given that their proposed classifier was designed to exclude tumors with marked immune and stromal cell infiltration – features that were considered non-pathognomonic of claudin-low tumors by the authors. It’ll be reassuring to know that the proposed classifier is not overly excluding immune-rich histological subtypes of breast cancer.

We would first like to emphasize that the proposed classifier was not designed to exclude tumors with marked stromal and immune infiltration. Rather, the classifier was designed such that stromal and immune infiltration would not be the deciding factor for claudin-low classification (discussed further in point vi and Reviewer #1 point 1D). Basal-like CoreCL tumors showed high levels of immune cell infiltration, as illustrated in Supplementary Fig. 5a.

We refer to the response to Reviewer #1 points 1A and 4 for a comprehensive evaluation of histological characteristics. Specifically, with regard to basal-like tumors in the METABRIC cohort, we find the following histological classifications for tumors with histological data available¹:

Histology	Basal-like CoreCL (n = 19)	Basal-like OtherCL (n = 20)	Basal-like (n = 205)
NST	15 (79%)	17 (85%)	181 (88%)
Lobular	1 (5%)	0	0
Mixed/NST	1 (5%)	0	5 (2%)
Medullary-like	1 (5%)	2 (10%)	12 (6%)
Tubular	0	0	0
Mucinous	0	0	0
Special types	1 (5%)	1 (5%)	7 (3%)

Sum of percentages may not equal 100 due to rounding

NST = No special type

While there are too few samples to draw firm conclusions, there do not appear to be major biases toward any histological subtypes in the basal-like tumors.

vi. High immune gene expression has been consistently described in the literature as a feature of claudin-low tumors. In fact, the authors cited this immune repertoire as being relevant for the efficacy of chemo and immunotherapies in claudin-low tumors in their discussion. I am curious to understand how they align this statement with the fact that their condensed claudin-low gene list excluded immune gene expression by focusing only on gene expression characteristics related to cell-cell adhesion, epithelial-mesenchymal transition, and stem cell-like/less differentiation.

The purpose of the condensed claudin-low gene list was to identify tumors with gene expression characteristics which are unique to claudin-low tumors. As the reviewer correctly states, high immune infiltration is an important feature of claudin-low tumors and is perhaps one of the more clinically relevant characteristics. However, immune and stromal infiltration are not unique to claudin-low tumors (see response to Reviewer #1 points 1A & 1C). Claudin-low tumors identified using the condensed gene list show high levels of immune infiltration (see Supplementary Figs. 5 & 7), but immune infiltrated tumors in general might more appropriately be identified using deconvolution methods such as ESTIMATE¹⁶. The condensed gene list could therefore be viewed as an attempt to reduce the overlap between a claudin-low predictor and an immune/stromal predictor (see response to Reviewer #1 point 1D for more detailed discussion).

vii. In the METABRIC cohort, the majority of Other-CL tumors were Basal-like with very little contribution from Luminal A and Normal-like. This was however different in the Oslo2 cohort where a substantial proportion of the Other-CL tumors were Luminal A. What could be responsible for these differences?

The reviewer raises an interesting question, which is challenging to answer conclusively. Several factors which may play a role include:

- *Chance: there are only 12 OtherCL tumors in the Oslo2 cohort, which are subdivided into the five intrinsic subtypes. This small sample size, relative to the number of groups, allows a minor amount of random variation to quite heavily influence the results.*
- *Selection bias: as we illustrate in Fig. 5, the variation in claudin-low prevalence between cohorts may be related to tumor purity cut-off. In Supplementary Fig. 3b, we can observe a slightly higher level of stromal infiltration in LumA claudin-low tumors than in basal-like claudin-low tumors. Perhaps the tumor purity cut-off in*

the METABRIC cohort disproportionately affects LumA tumors with marked stromal infiltration.

- *Composition of the cohort: Clinicopathological and/or demographic factors may have differed between the cohorts. Variations in studies (e.g. geography, year of recruitment, etc.) may cause slight skew in, for example, age groups, disease stages, prior treatment, ancestry, etc.*
- *Technical variations: There are minor technical variations between studies, such as RNA-isolation, choice of microarray technology, quality control procedures, data normalization, bioinformatical analysis/subtyping.*

In sum, there are many plausible factors which may have caused this difference between the two cohorts, none of which immediately strike us as a definitive answer. With only 12 OtherCL tumors in the Oslo2 cohort, we are hesitant to speculate too heavily on this matter.

Reviewer #3

This is a well written paper by Fougner and colleagues, in which they propose that Claudin-low breast cancer is not a unique subtype of breast cancer but rather an element or other dimension that can exist in breast cancers of all intrinsic subtypes albeit more prevalent in basal-like tumours.

Using an *in silico* approach they identify 87 claudin-low tumours in the Metabric data base using the 9 cell line classifier of Prat and colleagues. The majority (51.7%) of the claudin-low tumours so identified were basal, 32.2% normal-like and 10.3% luminal A. Then, through a series of comparisons they demonstrate that the claudin-low phenotype of each sub-group is reflective of their intrinsic subtype; including ER expression, lower mutational burden, lower genomic instability, driver mutations, proliferation etc.. In addition, they demonstrate that the survival of claudin-low tumours is not inferior to that of non-claudin low tumours of the same intrinsic subtype.

The authors then refine, again by *in silico* techniques the gene signature of claudin low tumours to a core 19 gene list and apply this new classifier to the Metabric database. This approach yielded 79 'core' claudin low tumours that are predominantly basal-like and yet have sustained and recognisable differences from the non-claudin low basal-like tumours including lower proliferation, lower mutational burden, increased genomic stability and increased stromal cell and immune cell infiltrate. This new classifier was subsequently validated in 2 different breast cancer databases.

The value of this work is that it does validate the existence of the claudin-low phenotype with some evidence to suggest its predominantly a group of basal-like breast cancers with immune infiltrate. The latter may have clinical implications.

We thank the reviewer for the favorable evaluation of our study.

Other changes

- 1) During the revision process, we found that we had misinterpreted the inclusion criteria for the METABRIC cohort. In the initial submission, we stated that there was a cellularity cut-off of 40% for the entire METABRIC cohort. Rather, the METABRIC cohort was originally divided into a discovery cohort and a validation cohort; the discovery cohort had a cellularity cut-off of 40%, but there was no cut-off for the validation cohort¹⁷. We have updated Fig. 5 and the associated text to reflect this. The updated Fig. 5 is as follows:

Sub-dividing the METABRIC cohort into the discovery (MB-Discov.) and validation (MB-Val.) datasets revealed a higher claudin-low prevalence in the validation cohort (5.6%) than in the discovery cohort (3.6%). Our argument that claudin-low prevalence may be influenced by cellularity cut-off was therefore reinforced by this correction.

- 2) We have moved the table showing sample numbers in the METABRIC cohort from the supplementary files to the main text (now Table 1), in order to make this information more accessible.

References

1. Mukherjee, A. *et al.* Associations between genomic stratification of breast cancer and centrally reviewed tumour pathology in the METABRIC cohort. *NPJ breast cancer* **4**, 5 (2018).
2. Aure, M. R. *et al.* Integrative clustering reveals a novel split in the luminal A subtype of breast cancer with impact on outcome. *Breast Cancer Res.* **19**, 44 (2017).
3. Williams, E. D., Gao, D., Redfern, A. & Thompson, E. W. Controversies around epithelial–mesenchymal plasticity in cancer metastasis. *Nat. Rev. Cancer* 1–17 (2019).
4. Weigelt, B. *et al.* Metaplastic breast carcinomas display genomic and transcriptomic heterogeneity. *Mod. Pathol.* **28**, 340 (2015).
5. Reis-Filho, J. S. *et al.* Metaplastic breast carcinomas are basal-like tumours.

- Histopathology* **49**, 10–21 (2006).
6. Morel, A. P. *et al.* A stemness-related ZEB1-MSRB3 axis governs cellular pliancy and breast cancer genome stability. *Nat. Med.* **23**, 568–578 (2017).
 7. Prat, A. *et al.* Phenotypic and molecular characterization of the claudin-low intrinsic subtype of breast cancer. *Breast Cancer Res.* **12**, R68 (2010).
 8. Bruna, A. *et al.* TGF β induces the formation of tumour-initiating cells in claudin low breast cancer. *Nat. Commun.* **3**, 1055 (2012).
 9. Morel, A.-P. *et al.* EMT inducers catalyze malignant transformation of mammary epithelial cells and drive tumorigenesis towards claudin-low tumors in transgenic mice. *PLoS Genet.* **8**, e1002723 (2012).
 10. Dias, K. *et al.* Claudin-low breast cancer; clinical & pathological characteristics. *PLoS One* **12**, e0168669 (2017).
 11. Chavez, K. J., Garimella, S. V & Lipkowitz, S. Triple negative breast cancer cell lines: one tool in the search for better treatment of triple negative breast cancer. *Breast Dis.* **32**, 35–48 (2010).
 12. Hanahan, D. & Coussens, L. M. Accessories to the crime: functions of cells recruited to the tumor microenvironment. *Cancer Cell* **21**, 309–322 (2012).
 13. Alsuliman, A. *et al.* Bidirectional crosstalk between PD-L1 expression and epithelial to mesenchymal transition: significance in claudin-low breast cancer cells. *Mol. Cancer* **14**, 149 (2015).
 14. Taylor, N. A. *et al.* Treg depletion potentiates checkpoint inhibition in claudin-low breast cancer. *J. Clin. Invest.* **127**, 3472–3483 (2017).
 15. Puisieux, A., Pommier, R. M., Morel, A.-P. & Laval, F. Cellular pliancy and the multistep process of tumorigenesis. *Cancer Cell* **33**, 164–172 (2018).
 16. Yoshihara, K. *et al.* Inferring tumour purity and stromal and immune cell admixture from expression data. *Nat. Commun.* **4**, 2612 (2013).
 17. Curtis, C. *et al.* The genomic and transcriptomic architecture of 2,000 breast tumours reveals novel subgroups. *Nature* **486**, 346 (2012).

Reviewers' comments:

Reviewer #1 (Remarks to the Author):

The authors have done a decent job addressing most of the prior concerns and adding new data such as the “startlingly strong correlation between stromal score (from ESTIMATE) and distance to the claudin-low centroid ($R^2 = 0.76$)”, that still suggest that the claudin predictor may not be able to discriminate normal stroma infiltration from claudin-low tumors. The authors arbitrary “setting a cut-off at a level such that 80% of claudin-low tumors have a StromalScore above the cut-off” and observation that four claudin-low tumors have a low stromal score does not sufficiently address the concern that the claudin-low predictor is not just measuring stroma. Given the high correlation between claudin-low and stroma and presence among all the intrinsic subtypes, perhaps the claudin-low predictor is just measure tumors of any subtype that have invaded the surrounding normal tissue and the gene expression is just a blend of that intrinsic tumor plus the amount of normal stroma in the sections. The authors were previously asked to evaluate the claudin-low subtype in readily available gene expression data from patient-matched normal adjacent tissue and tumor tissue from over 100 patients in the TCGA in prior comment 1C, but have not provided the requested analysis, hoping the that responses to 1A and 1D would suffice. However, the authors are missing an easy opportunity to shed more light on the claudin-low subtype by analyzing matched normal adjacent tissue available in the TCGA.

Major remaining concern from prior review:

Prior concern 1B. Also subtyping of paired tumor and adjacent normal tissue could reveal similar conclusions. There are several available in the TCGA and there are likely additional publicly available datasets on GEO or SRA. A quick analysis of TCGA that includes 1095 tumor and 113 normal adjacent provides interesting clues as the number of claudin-low tumors increases from 4% to 40% when only tumors are normalized together versus when tumor and adjacent normal samples are normalized together. Furthermore, greater than 80% of the matched normal are predicted as claudin-low and majority of the paired tumor is classified as other subtype. The authors should consider doing a more throughout analysis in the TCGA that include normal or normal adjacent breast and evaluating tumor cellularity in the comparisons.

Author Response:

The reviewer’s observations are intriguing, and from what we gather, the concern the reviewer is raising is that claudin-lowness (as measured by the nine-cell line predictor) might be heavily influenced by stromal infiltration. This is a vital consideration, and this point is directly demonstrated by the reviewer’s suggestion in point 1C. We also approach this question in 1A & 1D, as well as discussing the question at large in response to Reviewer #2, point i. We hope that these responses adequately address this concern.

Additional concerns relating to 1B:

It is unclear why the authors have decided to forego the recommended additional analysis that could shed more light on the confounding issue of normal stroma by evaluating patient-matched tumor and adjacent normal tissue. There are 113 breast cancers in the TCGA that have been RNA-seq normalized gene expression data available (<https://gdac.broadinstitute.org/>) that require no additional processing. Of these tumor samples (barcodes TCGA-XX-XXX-01), 113 have identical barcodes from matched normal tissue from the same patient (TCGA-XX-XXX-11). These samples also have tumor cellularity data available. The authors can use an R package (genefu) to determine the intrinsic subtypes of these samples with patient-matched normal tissue after the gene expression have been normalized between samples. They should perform two analysis, 1) just using the 113 tumor samples normalized together then intrinsic/claudin-low correlation and 2) using both the 113 tumor and the 113 matched normal samples normalized together prior to intrinsic/claudin-low prediction. The authors can then present a summary of the claudin-low subtype in normal tissue and in matched tumors. The authors should also annotate these samples by tumor cellularity. These data are available for download with Broad Firehose by selecting clinical

file "Merge_Clinical" and using the data from row "patient.samples.sample.portions.portion.slides.slide.percent_tumor_cells" and "patient.samples.sample.portions.portion.slides.slide.percent_stromal_cells". The inclusion of this additional analysis would provide much greater insight on the origins and limitation of the claudin-low predictor.

Reviewer #2 (Remarks to the Author):

The authors have offered a comprehensive response to the pertinent issues raised. To the extent that it was impractical in the current study, mainly due to limited sample size, the authors have provided further discussion on the characteristics of the claudin-low phenotype which were driven by normal vs tumor cell infiltration. In addition, the authors have included detailed discussion of the study's limitations as well as emphasizing the need for further studies, including single-cell transcriptomics, to conclusively answer the central questions. The authors have also provided detailed histology data to argue that observed differences may not be driven by tumor histology. A few outstanding concerns remain:

In the manuscript, as well the rebuttal letter, the authors did an excellent job in demonstrating that the Claudin-low phenotype is not a distinct subtype of breast cancer but a heterogeneous entity that can be present in other intrinsic subtypes. However, a fundamental issue that remains to be addressed pertains to the relevance of their proposed new classifier and how it materially departs from the nine cell-line classifier. Perhaps noteworthy is the fact that the major analyses that informed the main conclusion were based on the nine-cell line classifier, but the conclusions were not different when using the new classifier.

Apart from the basal-like tumors, the proposed new classifier did not result in substantial re-classification of claudin-low tumors within the other intrinsic subtypes. Although the authors' motivation for the new classifier was "in order to reduce a potentially confounding influence of normal cell infiltration and more accurately isolate features intrinsic to claudin-low tumors", it is not immediately evident if the classifier is achieving this end. For example, in response to Reviewer #1's comment (1C) the authors defined a StromalScore cut-off point guaranteeing 80% sensitivity to detect claudin-low tumors. However, 34% of the tumors were classified as CoreCL in contrast to 24% that were classified as Claudin-low when using this cutoff-point. This further begs the question of whether this new classifier is, in fact, limiting the confounding influence of normal cell infiltration beyond what is obtained with the nine-cell classifier.

Finally, the authors state in one of their responses that: "ultimately, we can only evaluate how good a classifier is in the context of the question: Good for what? Claudin-low status does not appear to influence survival, and we can therefore not evaluate a classifier by its prognostic value. There are no currently available targeted therapies against claudin-low features, and we can therefore not evaluate a classifier for its ability to predict treatment response. There are no biological gold standards defining claudin-low, such as histology or protein markers, which a classifier could be benchmarked against. Therefore, we cannot claim that our method is "better" than the nine-cell line predictor. Rather, we present evidence that our method is more appropriate for identifying tumors defined by EMT-like gene expression patterns and genomic stability."

While agreeing with the views expressed by the authors above, another question that needs to be asked is: what is the value of a claudin-low classifier that more appropriately identifies tumors defined by EMT-like gene expression patterns and genomic stability? Although this may be obvious to the authors, it is not immediately evident throughout the manuscript's discussion.

We thank the reviewers for the continued insightful comments. Please find responses to the concerns below.

Reviewer #1 (Remarks to the Author):

The authors have done a decent job addressing most of the prior concerns and adding new data such as the “startlingly strong correlation between stromal score (from ESTIMATE) and distance to the claudin-low centroid ($R^2 = 0.76$)”, that still suggest that the claudin predictor may not be able to discriminate normal stroma infiltration from claudin-low tumors. The authors arbitrary “setting a cut- off at a level such that 80% of claudin-low tumors have a StromalScore above the cut- off” and observation that four claudin-low tumors have a low stromal score does not sufficiently address the concern that the claudin-low predictor is not just measuring stroma. Given the high correlation between claudin-low and stroma and presence among all the intrinsic subtypes, perhaps the claudin-low predictor is just measure tumors of any subtype that have invaded the surrounding normal tissue and the gene expression is just a blend of that intrinsic tumor plus the amount of normal stroma in the sections. The authors were previously asked to evaluate the claudin-low subtype in readily available gene expression data from patient-matched normal adjacent tissue and tumor tissue from over 100 patients in the TCGA in prior comment 1C, but have not provided the requested analysis, hoping the that responses to 1A and 1D would suffice. However, the authors are missing an easy opportunity to shed more light on the claudin-low subtype by analyzing matched normal adjacent tissue available in the TCGA.

Major remaining concern from prior review:

Prior concern 1B. Also subtyping of paired tumor and adjacent normal tissue could reveal similar conclusions. There are several available in the TCGA and there are likely additional publicly available datasets on GEO or SRA. A quick analysis of TCGA that includes 1095 tumor and 113 normal adjacent provides interesting clues as the number of claudin-low tumors increases from 4% to 40% when only tumors are normalized together versus when tumor and adjacent normal samples are normalized together. Furthermore, greater than 80% of the matched normal are predicted as claudin-low and majority of the paired tumor is classified as other subtype. The authors should consider doing a more throughout analysis in the TCGA that include normal or normal adjacent breast and evaluating tumor cellularity in the comparisons.

Author Response:

The reviewer’s observations are intriguing, and from what we gather, the concern the reviewer is raising is that claudin-lowness (as measured by the nine-cell line predictor) might be heavily influenced by stromal infiltration. This is a vital consideration, and this point is directly demonstrated by the reviewer’s suggestion in point 1C. We also approach this

question in 1A & 1D, as well as discussing the question at large in response to Reviewer #2, point i. We hope that these responses adequately address this concern.

Additional concerns relating to 1B:

It is unclear why the authors have decided to forego the recommended additional analysis that could shed more light on the confounding issue of normal stroma by evaluating patient-matched tumor and adjacent normal tissue. There are 113 breast cancers in the TCGA that have been RNA-seq normalized gene expression data available (<https://gdac.broadinstitute.org/>) that require no additional processing. Of these tumor samples (barcodes TCGA-XX-XXX-01), 113 have identical barcodes from matched normal tissue from the same patient (TCGA-XX-XXX-11). These samples also have tumor cellularity data available. The authors can use an R package (genefu) to determine the intrinsic subtypes of these samples with patient-matched normal tissue after the gene expression have been normalized between samples. They should perform two analysis, 1) just using the 113 tumor samples normalized together then intrinsic/claudin-low correlation and 2) using both the 113 tumor and the 113 matched normal samples normalized together prior to intrinsic/claudin-low prediction. The authors can then present a summary of the claudin-low subtype in normal tissue and in matched tumors. The authors should also annotate these samples by tumor cellularity. These data are available for download with Broad Firehose by selecting clinical file "Merge_Clinical" and using the data from row "patient.samples.sample.portions.portion.slides.slide.percent_tumor_cells" and "patient.samples.sample.portions.portion.slides.slide.percent_stromal_cells". The inclusion of this additional analysis would provide much greater insight on the origins and limitation of the claudin-low predictor.

We are grateful to the reviewer for constructive remarks and detailed suggestions for further analysis. We have performed these analyses, as we have understood them, using the file "illuminahisec_rnaseq2-RSEM_genes" from the website linked above. This file contained 112 tumor adjacent samples with an -11 barcode, and 1093 primary tumor samples with an -01 barcode. The following analyses were carried out:

UQ-Normalize	Scale & Center	Classify
1093 tumors & 112 T.A.N.	112 tumors with 112 matched T.A.N.	9CL & PAM50
1093 tumors & 112 T.A.N.	112 tumors	9CL & PAM50
1093 tumors	112 tumors	9CL & PAM50
112 tumors	112 tumors	9CL & PAM50
112 tumors & 112 T.A.N.	112 tumors with 112 matched T.A.N.	9CL & PAM50
112 tumors & 112 T.A.N.	112 tumors	9CL & PAM50

T.A.N. = Tumor adjacent normal

UQ = Upper quartile

9CL = Nine-cell line claudin-low predictor

112 tumors = Tumors for which matched T.A.N. sample was available

The results from the classification step were not affected by which samples were UQ-normalized together.

When only tumors were scaled and centered together, 5 of the 112 tumors were classified as claudin-low using the nine-cell line predictor. 4 of those 5 tumors were also classified as claudin-low in our original analyses of the entire TCGA-BRCA dataset (described in the manuscript). One tumor was classified as claudin-low when the entire cohort was analyzed together, but not when only the 112 tumors were analyzed.

When the tumors were scaled and centered together with T.A.N. samples, no tumor samples were classified as claudin-low. 21 of 112 T.A.N. samples were classified as claudin-low. Claudin-low-like features in normal mammary gland samples have previously been reported by ourselves and colleagues^{1,2}.

In sum, our findings show that scaling/centering of tumor samples together with T.A.N. samples reduces the number of tumors classified as claudin-low. Since tumor tissue and T.A.N. tissue are inherently different, gene centering may create an artificial skewing of the data. A subset of normal breast tissue samples show claudin-low-like/mesenchymal features^{1,2}, and centering T.A.N. together with tumor can therefore skew tumor gene expression away from the EMT-like features. This might explain why fewer tumors are classified as claudin-low when scaled/centered with T.A.N.

Our findings contrast with the reviewer's suggestion that normalizing tumor and T.A.N. together leads to a drastic increase in the number of tumors classified as claudin-low. We hope our interpretation of the suggested analyses is in line with the reviewer's request. The script used for these analyses is enclosed with the re-submission along with a summary of the output.

The five claudin-low tumors identified in the above analyses had tumor cell percentages of: 40, 65, 70, 70 and 100 (mean: 69%; mean for all 112 tumors: 73%). The corresponding stromal cell percentages were: 60, 33, 24, 30 and 0 (mean: 29%, mean

for all 112 tumors: 19%). These findings mirror those described in our previous response to reviewer concerns: claudin-low tumors generally have higher degrees of stromal infiltration, but only a minority of tumors with high levels of stromal infiltration are classified as claudin-low. For example, in the 112 tumors analyzed, 12 tumors have a stromal infiltration $\geq 40\%$, of which only one is classified as claudin-low.

With respect to the intrinsic subtypes, predicted using PAM50, the results were as follows:

Intrinsic subtype	Tumor Scaled and centered alone	Tumor Scaled and centered with T.A.N.	T.A.N Scaled and centered with tumor
Basal-like	17	16	6
HER2E	12	26	0
LumA	42	20	3
LumB	38	49	0
Normal-like	3	1	103

Scaling and centering tumor samples together with T.A.N. primarily shifted the subtypes of many LumA samples to HER2E and LumB. Most T.A.N. samples were classified as normal-like. While few tumor samples were classified as normal-like, it is interesting to note that 2 of 3 normal-like tumors are no longer classified as such when scaled together with T.A.N. samples. This also illustrates that scaling tumor tissue together with normal tissue skews tumor gene expression away from normal breast tissue-like features.

Regarding the broader question of whether the nine-cell line predictor is able to distinguish “genuine” claudin-low tumors from non-claudin-low tumors with stromal infiltration, we would like to emphasize that we largely agree with the reviewer’s concern. Identifying limitations in the nine-cell line predictor is an important theme in our study, which we note, for example, in lines 23-25, 68-70, 231-233, 243-245, and 290-292. With respect to the question of whether claudin-low truly exists or is an artefact of stromal infiltration, we refer to lines 325-345 in the manuscript, and the response to Reviewer #2, point **i** in the previous review round. Claudin-low only being an artefact of stromal infiltration is implausible in light of these lines of evidence, but we acknowledge that the possibility cannot be formally excluded without single-cell analyses (lines 360-364).

The results from our analyses are incongruent with the reviewer’s anticipated results, and have not materially affected our conclusions. We have therefore chosen not to

include them in the revised manuscript. We have added further emphasis of the similarities between claudin-low gene expression patterns and normal breast tissue (lines 328 – 329). We also note that this topic is discussed in the context of the limitations of our study in lines 360-372.

Reviewer #2 (Remarks to the Author):

The authors have offered a comprehensive response to the pertinent issues raised. To the extent that it was impractical in the current study, mainly due to limited sample size, the authors have provided further discussion on the characteristics of the claudin-low phenotype which were driven by normal vs tumor cell infiltration. In addition, the authors have included detailed discussion of the study's limitations as well as emphasizing the need for further studies, including single-cell transcriptomics, to conclusively answer the central questions. The authors have also provided detailed histology data to argue that observed differences may not be driven by tumor histology.

A few outstanding concerns remain:

In the manuscript, as well the rebuttal letter, the authors did an excellent job in demonstrating that the Claudin-low phenotype is not a distinct subtype of breast cancer but a heterogenous entity that can be present in other intrinsic subtypes. However, a fundamental issue that remains to be addressed pertains to the relevance of their proposed new classifier and how it materially departs from the nine cell-line classifier. Perhaps noteworthy is the fact that the major analyses that informed the main conclusion were based on the nine-cell line classifier, but the conclusions were not different when using the new classifier.

Apart from the basal-like tumors, the proposed new classifier did not result in substantial re-classification of claudin-low tumors within the other intrinsic subtypes. Although the authors' motivation for the new classifier was "in order to reduce a potentially confounding influence of normal cell infiltration and more accurately isolate features intrinsic to claudin-low tumors", it is not immediately evident if the classifier is achieving this end. For example, in response to Reviewer #1's comment (1C) the authors defined a StromalScore cut-off point guaranteeing 80% sensitivity to detect claudin-low tumors. However, 34% of the tumors were classified as CoreCL in contrast to 24% that were classified as Claudin-low when using this cutoff-point. This further begs the question of whether this new classifier is, in fact, limiting the confounding influence of normal cell infiltration beyond what is obtained with the nine-cell classifier.

Finally, the authors state in one of their responses that: "ultimately, we can only evaluate how good a classifier is in the context of the question: Good for what? Claudin-low status does not appear to influence survival, and we can therefore not evaluate a classifier by its prognostic value. There are no currently available targeted therapies against claudin-low features, and

we can therefore not evaluate a classifier for its ability to predict treatment response. There are no biological gold standards defining claudin-low, such as histology or protein markers, which a classifier could be benchmarked against. Therefore, we cannot claim that our method is “better” than the nine-cell line predictor. Rather, we present evidence that our method is more appropriate for identifying tumors defined by EMT-like gene expression patterns and genomic stability.”

While agreeing with the views expressed by the authors above, another question that needs to be asked is: what is the value of a claudin-low classifier that more appropriately identifies tumors defined by EMT-like gene expression patterns and genomic stability? Although this may be obvious to the authors, it is not immediately evident throughout the manuscript’s discussion.

We thank the reviewer for constructive comments and suggestions for further improvements. In order to address these concerns and clarify our reasoning, we would first like to discuss cellular plasticity in tumorigenesis³. Generally, more highly differentiated tumors are more committed to a defined cellular lineage, and an increased number of genetic perturbations are required for a cell to undergo malignant transformation. Less differentiated tumors are less committed to a defined lineage, and fewer perturbations will be required for malignant transformation. Importantly, differentiated cells may be transdifferentiated through EMT - induced by microenvironmental signaling - which reduces the number of genetic perturbations required for malignant transformation³⁻⁵. This concept was studied by Morel et al. in triple negative breast tumors, which revealed that the EMT-transcription factor ZEB1 promotes malignant transformation while maintaining genomic stability⁵. Claudin-low tumors showed high expression of ZEB1 and other EMT-related genes, and associated paucity of copy number aberrations. We consider these studies to provide the most compelling etiological explanation for the claudin-low phenotype.

In light of these advances made since claudin-low tumors were first characterized, we determined the following to be the most pertinent criteria for evaluating claudin-low/CoreCL tumors:

- *Genomic stability (both CNAs and mutations)*
- *IntClust4 subtype*
- *EMT-like gene expression features (in conjunction with the other pathognomonic claudin-low gene expression features)*

We have put forth considerable evidence that these features are more strongly selected for in CoreCL tumors than in claudin-low tumors identified by the nine-cell line predictor:

- Figs. 3 & 4
- Comparison of Fig.1 and Supplementary Fig. 7
- Supplementary Figs. 4 – 6
- Lines 256 – 258 and 263-266
- The second set of figures in response to Reviewer #1, point 1C, in the previous round of reviews

Microenvironmental signaling may be the factor that induces EMT in claudin-low tumor (initiating) cells³⁻⁵. We do therefore not necessarily view a higher StromalScore in CoreCL tumors than in nine-cell line claudin-low tumors as a weakness of the method. Nonetheless, we have removed the noted phrase (“reduce the confounding effect...”, line 188) in order to tone down our assertions.

Regarding the question of why it is more valuable to select for CoreCL tumor features than nine-cell line claudin-low tumor features, we would argue that both classification methods are in fact attempting to identify the same tumor population. However, the tumors identified as claudin-low by the nine-cell line predictor contain an admixture of tumors which do not seem to share the etiological basis which defines “genuine” claudin-low tumors. The goal of these two methods is therefore the same, but the aforementioned lines of evidence indicate that the method presented in our study achieves this goal more accurately.

Finally, we would again like to acknowledge that our method has certain limitations, and re-iterate that we do not view our method as an ultimate solution to classifying claudin-low tumors. It is therefore important to note that we do not present our method as such in the manuscript; we primarily use it to glean new understanding of the claudin-low phenotype, and to clarify limitations in the established method.

We have updated the following lines to tone down claims regarding our method for identifying claudin-low tumors: 23, 68, 188, 289, 309 and 374-375. We have also added mention of certain considerations discussed above in lines 42-44, 324 and 341-345.

References

1. Haakensen, V. D. *et al.* Gene expression profiles of breast biopsies from healthy women identify a group with claudin-low features. *BMC Med. Genomics* **4**, 77 (2011).
2. Bergholtz, H. *et al.* A Longitudinal Study of the Association between Mammographic Density and Gene Expression in Normal Breast Tissue. *J. Mammary Gland Biol. Neoplasia* 1–13 (2019).
3. Puisieux, A., Pommier, R. M., Morel, A.-P. & Lavial, F. Cellular pliancy and the multistep process of tumorigenesis. *Cancer Cell* **33**, 164–172 (2018).
4. Morel, A.-P. *et al.* EMT inducers catalyze malignant transformation of mammary epithelial cells and drive tumorigenesis towards claudin-low tumors in transgenic mice. *PLoS Genet.* **8**, e1002723 (2012).

5. Morel, A. P. *et al.* A stemness-related ZEB1-MSRB3 axis governs cellular pliancy and breast cancer genome stability. *Nat. Med.* **23**, 568–578 (2017).

REVIEWERS' COMMENTS:

Reviewer #1 (Remarks to the Author):

The additional analysis performed by the authors is appreciated and have alleviated all my concerns.

Reviewer #2 (Remarks to the Author):

thank you or your thorough answers
The manuscript is markedly better

Reviewer #1 (Remarks to the Author):

The additional analysis performed by the authors is appreciated and have alleviated all my concerns.

Reviewer #2 (Remarks to the Author):

Thank you for your thorough answers. The manuscript is markedly better.

We thank the reviewers for the time and effort spent in evaluating our manuscript. The suggested analyses and amendments were insightful and improved the quality of the study.